## Overview Review

Well-being; mental health; intervention; ethnic minority; migration; review; Europe; cultural adaptation; community participation

**Corresponding author:**
Hanne Apers;
Email: Hanne.Apers@UAntwerpen.be

# Interventions to improve the mental health or mental well-being of migrants and ethnic minority groups in Europe: A scoping review

Hanne Apers[1] ⓘ, Lore Van Praag[2] ⓘ, Christiana Nöstlinger[3] ⓘ and Charles Agyemang[4] ⓘ

[1]Centre for Migration and Intercultural Studies/Centre for Population, Family and Health, University of Antwerp, Antwerp, Belgium; [2]Erasmus School of Social and Behavioural Sciences, Erasmus University Rotterdam, Rotterdam, Netherlands; [3]Department of Public Health, Institute of Tropical Medicine Antwerp, Antwerp, Belgium and [4]Department of Public and Occupational Health, Amsterdam Public Health Research Institute, Amsterdam UMC, University of Amsterdam, Amsterdam, Netherlands

## Abstract

In Europe, migrants and ethnic minority groups are at greater risk for mental disorders compared to the general population. However, little is known about which interventions improve their mental health and well-being and about their underlying mechanisms that reduce existing mental health inequities. To fill this gap, the aim of this scoping review was to synthesise the available evidence on health promotion, prevention, and non-medical treatment interventions targeting migrants and ethnic minority populations. By mapping and synthesising the findings, including facilitators and barriers for intervention uptake, this scoping review provides valuable insights for developing future interventions. We used the PICo strategy and PRISMA guidelines to select peer-reviewed articles assessing studies on interventions. In total, we included 27 studies and synthesised the results based on the type of intervention, intervention mechanisms and outcomes, and barriers and facilitators to intervention uptake. We found that the selected studies implemented tailored interventions to reach these specific populations who are at risk due to structural inequities such as discrimination and racism, stigma associated with mental health, language barriers, and problems in accessing health care. The majority of interventions showed a positive effect on participants' mental health, indicating the importance of using a tailored approach. We identified three main successful mechanisms for intervention development and implementation: a sound theory-base, systematic adaption to make interventions culturally sensitive and participatory approaches. Moreover, this review indicates the need to holistically address social determinants of health through intersectoral programming to promote and improve mental health among migrants and ethnic minority populations. We identified current shortcomings and knowledge gaps within this field: rigorous intervention studies were scarce, there was a large diversity regarding migrant population groups and few studies evaluated the interventions' (cost-)effectiveness.

## Impact statement

This scoping review highlights the importance of tailored interventions for migrants and ethnic minorities confronted with specific mental health challenges. Our findings provide valuable insight for healthcare policy-makers, local governments and scientific experts. Studies report high prevalence of mental disorders among these population groups. At the same time, structural and sociocultural barriers limit their access to mental healthcare and prevention. Therefore, the need for tailored interventions is high. This scoping review analysed 27 selected studies to identify what works in successfully promoting the mental well-being, and in preventing or dealing with mental health problems among migrants and ethnic minorities. This is the first comprehensive review that also identifies underlying intervention mechanisms leading to effective outcomes: (1) using a sound theory base, (2) systematically adapting the intervention in a culturally sensitive manner, and (3) the use of participatory approaches, preferably early on in their development. Our findings also point to the importance of holistic approaches addressing social health determinants to reduce mental health inequities. The findings and conclusions from this research should be used as guidance for the development of mental health interventions for diverse groups of migrants and ethnic minorities to make them more effective and sustainable.

## Introduction

Migrant and ethnic minority populations are at greater risk for mental health problems than the general population in Europe and the European Economic Area (EU/EEA) (Carta et al., 2005;

Fazel et al., 2005; Missinne and Bracke, 2012; Ekeberg and Abebe, 2021; Purgato et al., 2021). While it has been suggested that some groups of migrants in certain contexts may have a health advantage over nationals upon their arrival (Dhadda and Greene, 2018), this "healthy-migrant effect" appears to vanish with longer duration of residence due to increasing health inequalities. Evidence for migrants' decline in mental health over the years is convincing (Elshahat et al., 2021). Several studies show a higher prevalence of mental disorders such as post-traumatic stress disorder (PTSD), anxiety and depression among these populations, as well as of substance abuse and severe mental illnesses, such as psychosis in comparison with the majority population in the countries of residence (Fazel et al., 2005; Missinne and Bracke, 2012; Nosè et al., 2017; Turrini et al., 2017; Foo et al., 2018; Hynie, 2018; Ekeberg and Abebe, 2021).

Migration trajectories and integration processes tend to be a psycho-social process of loss and change, associated with several mental stressors and suffering (Bhugra, 2004; Carta et al., 2005; Derr, 2016). Migration drivers such as poverty, violent political conflicts, and climate-related disasters will continue to trigger global migration (O'Malley, 2018). The complex and interrelated combination of social and structural determinants pre-, during and post-migration impact migrants' mental health (International Organization for Migration, 2006; Spallek et al., 2011; World Health Organization, 2022). Difficult socio-economic circumstances in their countries of origin such as limited access to education, employment and healthcare, economic disruptions, individual or family-related stressors might have affected their health status prior to and upon migration (Davies et al., 2010; Priebe et al., 2016). Migrants may face many challenges before and during their migration trajectory: some migrant groups are exposed to violence and trauma, often in the form of human rights violations (Priebe et al., 2016; Lindert et al., 2008; Purgato et al., 2021). After arrival, resettlement stressors, such as difficult socio-economic and living circumstances, complex legal residence procedures, detention procedures, and experiences of discrimination and racism, among others, may negatively affect their mental health (Priebe et al., 2016; Lindert et al., 2008; Nosè et al., 2017; Von Werthern et al., 2018).

Similar mental health vulnerabilities have been observed among ethnic minorities born in European countries (Myers, 2009; Spallek et al., 2011; Borrell et al., 2015; Ikram, 2016; Hynie, 2018). The social determinants that impact migrant's health before, during and after migration may also affect their offspring and subsequent generations (Spallek et al., 2011). Different genetic factors, cultural beliefs and health behaviours persist over generations, and the socio-economic conditions of parents can determine the health situation of their children (Spallek et al., 2011). Migrant descendants show a greater likelihood of developing mental disorders such as PTSD, as trauma can be transmitted to later generations through psychosocial mechanisms within the parent–child attachment and intra-family communication style (Sangalang and Vang, 2017; Silwal et al., 2019). The complex issue of trauma transmission is not limited to family ties. Also, indirect experiences of racial discrimination, racial profiling, and racism were shown to affect the mental well-being among some ethnic minority groups (Cénat, 2020).

The evidence on the particular causes and circumstances of migrant groups' heightened vulnerabilities to ill mental health, gives reason for specific, targeted interventions on mental health promotion, prevention, and treatment, apart from interventions targeting the general non-migrant and/or ethnic majority population (Uphoff et al., 2020). Migrants and ethnic minorities might experience language, cultural and structural barriers that complicate access to regular mental healthcare (Uphoff et al., 2020) and thus they may make less use of health care services or use services in a different manner (Graetz et al., 2017). Most of the target-group specific interventions on mental health, however, seem to be directed to the specific subgroup of refugees and asylum seekers, who have specific needs given their specific migration history, distinct legal status, and access to health systems (Nosè et al., 2017; Lebano et al., 2020; Uphoff et al., 2020; Purgato et al., 2021). An overview of interventions focusing on refugees and asylum seekers can be found in the Cochrane Library (Uphoff et al., 2020) and further in this special issue to be published in the journal. Other migrant groups, such as economic migrants, as well as ethnic minorities who are subject to similar mental health risks, are not considered in those reviews.

Additionally, intervention reviews rather focus on those populations already diagnosed with a mental health condition and little emphasis is put on the prevention of mental health problems or promotion of mental well-being in those groups at increased risk (Purgato et al., 2021). However, prevention strategies and mental health promotion approaches are essential to ensure psychological well-being, reduce the mental health burden as well as to improve the mental health outcomes of migrant and ethnic minority groups (Foo et al., 2018). While it is clear that migrants and ethnic minorities are exposed to various risk factors, this exposure does not necessarily lead to the development of mental health problems. Resilience factors, such as social support, positive coping strategies, and personal characteristics can help individuals navigate adversity and prevent mental health problems to develop (Dubus, 2022). Resilience can also be fostered through utilising available resources to address mental health concerns. Therefore, interventions that focus on increasing resilience, such as by strengthening social networks, may help to prevent mental health problems among migrants and ethnic minorities.

In the current literature, no review could be found on mental health interventions for the broader group of migrants and ethnic minority populations in Europe. The existing review studies on refugees and asylum seekers are particularly relevant to shed light on these groups' specific needs, recognising the fact that forced migration may constitutes the highest mental health risk (Uphoff et al., 2020). However, we also need a better understanding of what renders mental health interventions effective for the larger group of migrants and ethnic minorities. This includes a broad range of people such as first-generation migrants (which may or may not include previous refugee experiences), second-generation migrants and ethnic minorities. Recognising the fact that no universally accepted definition of migrants exists (IOM, 2023), the current scoping review uses the International Organisation for Migration definition of migrants (IOM, 2019) "as anyone who moves away from their usual place of residence regardless of legal status, the reason for migration and the length of stay." This review aimed to fill the above-mentioned knowledge gaps for this broader groups of migrants and ethnic minorities as population of interest by mapping and synthesising the available evidence on effective approaches and interventions to improve their mental health and well-being.

## Methodology

A scoping review methodology was fit for the purpose given the broad field of inquiry and the likely mix of outcomes and research

designs adopted. We applied Arksey and O'Malley's multistage methodological framework (Arksey and O'Malley, 2005), taking into account Levac et al.'s (2010) refinements. The stages are: (1) Clarifying and linking the purpose and research question, (2) Identifying relevant studies and balancing feasibility with the comprehensiveness of the scoping process, (3) Applying an iterative team approach in the study selection, (4) Charting the data, and (5) collating and summarising the results through a qualitative thematic analysis and reporting implications of the findings for policy, practice and research. The systematic data selection was based on the PRISMA extension guidelines for scoping reviews (Tricco et al., 2018).

### Stage 1: Clarifying and linking the purpose of the review to the research question

#### Definitions

The term "Migrants and ethnic minorities" describes heterogeneous groups with numerous definitions. For this study, we used the definitions of the International Organization for Migration (IOM) (2019). In this framework, "migrant" is defined as "*an umbrella term, not defined under international law, reflecting the common lay understanding of a person who moves away from his or her place of usual residence, whether within a country or across an international border, temporarily or permanently, and for a variety of reasons.*" "Ethnic minority groups," are in this scoping review broadly defined as a group within a community that has a specific way of life, based on meanings, crucial for processes of identification and differentiation (Jenkins, 2008), which differs from the rest of the population. In many cases, but not always, ethnicity is intertwined with migration, increasing their significance and salience, which makes it also interesting to discuss together in this review (Erel et al., 2016).

To inform the further scoping process, we delineated the scope of the search, operationalised the search terms and defined a clear research question (Levac et al., 2010). During a first team meeting, we discussed and decided on the research question based on a PICo approach (Population, Interest and Context) (Stern et al., 2014; Eriksen and Frandsen, 2018; see Table 1). The protocol has been

registered at the Center for Open Science (https://doi.org/10.17605/OSF.IO/R8SBF).

The conceptualization of these terms led to a clearly articulated scope of inquiry and enabled us to develop the following research question for the review:

> What is known about interventions applied in the EU/EEA +UK to improve migrants' and ethnic minorities' mental health or well-being?

Building further on this general review question, we defined the following specific objectives: (1) To identify what interventions are available and their respective outcomes; (2) To provide an overview of the intervention mechanisms and culturally adapted delivery strategies applied within the selected studies, focusing on the specific target groups; and (3) To identify barriers and facilitators for intervention uptake. Finally, the goal of this scoping review is to give recommendations for policy and practice based on the critical appraisal of the available evidence.

### Stage 2: Identifying relevant studies

We added eligibility criteria for the search strategy to the PICo-criteria, such as year of publication, availability of text and language of publication. We searched for peer-review journal articles available in full-text and written in English until the date of the search, that is, 01/07/2022. We included all study designs. We excluded comments, letters to the editor, books and book chapters, conference abstracts and theses. We defined the search terms and potential databases, based upon available resources by the review team (i.e., consisting of the first three authors). The first author conducted a few try-out literature searches using different databases to check for the most suited search terms and date range. Interim results were continuously discussed by the team to develop the final, comprehensive search string. We searched three main relevant bibliographic databases from their inception, that is, PubMed, Web of Science and PsycInfo. Key search terms were a combination of the core concepts of our research objectives and related terms or synonyms. The core concepts were "intervention," "improving," "mental health," "migrants and ethnic minorities" and "countries

**Table 1.** Delineating the review question and refining the search strategy based on PICo-approach

| PICo search strategy | Inclusion criteria | Exclusion criteria |
|---|---|---|
| Population | Migrants (as defined by IOM) and Ethnic Minority groups. This includes, for example, second-generation migrants, economic migrants, people migrating because of family reunification, undocumented migrants<br>(While these definitions include the refugees and asylum seekers, we excluded them if interventions were solely targeted at refugees and asylum seekers, see exclusion criteria) | Studies solely focused on refugees and asylum seekers. *Rationale*: asylum seekers and refugees have different mental health needs. When ethnicity/migrant background is included as a control variable and not the focus of the study, we do not include them in the review |
| | Adults aged +18 years old | Children or adolescents aged −18 years old. *Rationale*: children and adolescents may experience different mental health needs and specified intervention approaches |
| Interest | Studies describing interventions oriented at improving mental health/well-being outcomes; other interventions can be included if they provide a clear link to mental health outcomes | Studies describing interventions without (a link to) outcome measures for mental health or well-being. Studies describing purely medical/pharmacological interventions are excluded. Articles relating to broad policies are excluded from the results, but we included them where relevant in the discussion to stimulate debate |
| Context | All countries of EU/EEA + UK and Switzerland; if more countries are involved in the study (e.g., high-income countries), we include the paper but only focus on the European part | Studies solely focused on countries outside EU/EEA + UK. *Rationale:* we only include studies related to the geographical region of Europe to limit the variability in study contexts |

from EU/EEA + UK." The fully applied search strings can be found in the Supplementary Material.

### Stage 3: Study selection

All results of the final literature searches were deduplicated using EndNote and were listed by the first author in a Microsoft Excel file. The first three authors screened the results (title and abstract) independently. During several team meetings, they discussed all results and selected relevant studies for full-text reading according to the eligibility criteria (as listed in Table 1). Reasons for exclusion after full-text reading were reported and categorised. If full-text reading revealed reference to other relevant articles, not included in the search results, an additional manual search was performed to include and assess those studies. Furthermore, we have scoped the individual studies within the meta-analyses. Most studies were also retrieved by our own search strategy. However, studies that met our inclusion criteria, were additionally included (e.g., Jacob et al., 2002; Chaudhry et al., 2009). Furthermore, we included the meta-analyses as well, as we deemed their analyses and comprehensive conclusions relevant for answering our review questions and the discussion of the results. The flowchart in Figure 1 illustrates the details of the search and selection process.

### Stage 4: Charting the data

The first author drafted a data extraction sheet using Microsoft Excel, which was piloted by the team with several articles. The data extraction sheet compiled the following key characteristics of the full-text articles: author, year of publication, title, country of study, participant characteristics, sample size (if applicable), methodological approach and study design, context of study, phenomenon of interest, theory of change (if applicable), intervention description, used instruments to measure outcomes, results and outcomes, barriers, facilitators, recommendations, study limitations, and data screeners' remarks. Reasons for exclusion after full-text reading were recorded. All selected articles were randomly assigned to one

of the three first authors to extract data, and cross-checked and discussed with the other team members. The categories used in the data extraction sheet form the basis for the next steps, that is, the synthesis of the results.

### Stage 5: Collating and summarising the results

We performed a thematic narrative synthesis of the selected articles to analyse the relevant thematic, methodological, and population-specific characteristics. We first inductively coded the intervention relating to their content (see the descriptive part under results). We then mapped the identified interventions along a continuum of disease prevention (i.e., primary prevention and promotion of well-being to prevent problems before they emerge) to secondary prevention (i.e., targeted interventions for people at high risk of developing mental disorders when exposed to specific risk factors) to tertiary prevention (i.e., focusing on interventions for people with acute or chronic mental health problems). The latter is distinct from pharmacological treatment (see eligibility criteria) but focusing on strategies to support patients in coping and living well with ill mental health including self-management. While we acknowledge that these stages may overlap and fluctuate in real life (Purgato et al., 2021), we use them for theoretical conceptualisation and because many health professionals are familiar with it. Clearly, this categorisation remains descriptive, and does not address underlying health disparities relevant for migrant mental health based on, for example, ethnic inequities or socio-economic status, as indicated by Compton and Shim (2020). Instead, Compton and Shim (2020) propose to look at how to reduce the population burden of social mental health determinants, which are highly interconnected. A true classification system for mental health interventions based on social determinants of health does not exist. We therefore aimed to contribute to this theoretical gap by analysing the intervention mechanisms in terms of addressing the different levels of social health determinants.

To provide accurate answers to the research objectives, we also analysed the intervention approaches used in-depth. For instance,

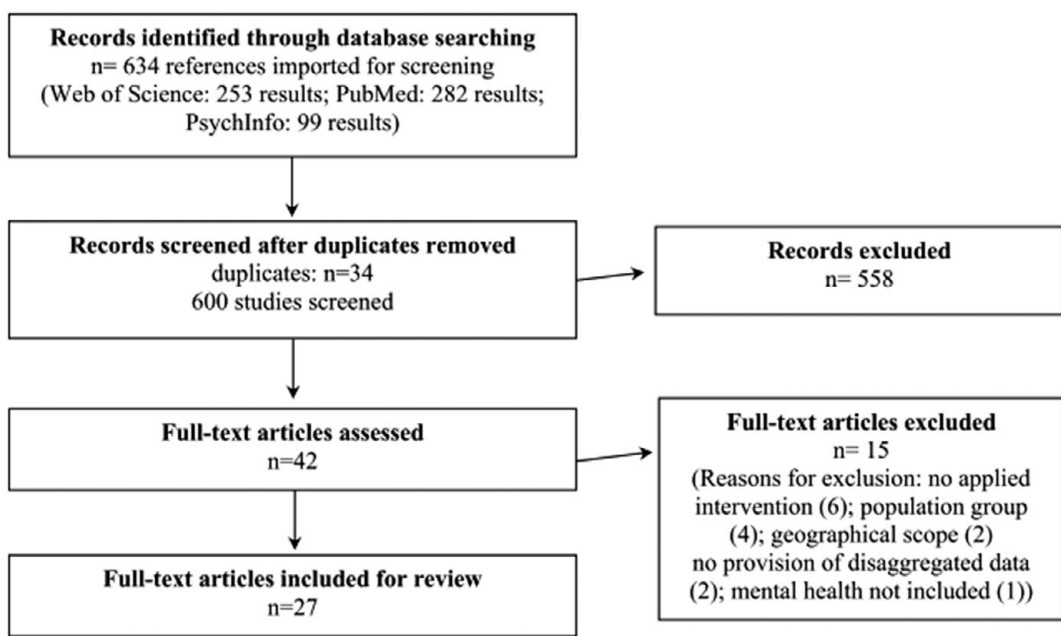

**Figure 1.** Flowchart of search and selection process.

we analysed the category "intervention description" with a specific focus on the interventions' cultural adaptation. Similarly, interventions describing a participatory approach, were labelled under different categories along the continuum proposed by Attygalle (2020): from community-informed (CI) over community-shaped (CS) interventions to community-driven (CD) initiatives. Interventions aiming to increase access to services at the respective stages of the prevention continuum were also included.

All authors commented on all results, conclusions drawn, and policy recommendations made and achieved consensus through discussion.

## Results

### Description of selected studies

We retrieved 282 results in PubMed starting from 1989, 253 results in Web of Science from 1993 and 99 results in PsycInfo from 2012. After the removal of duplicates and the screening process, we included 27 articles in the final selection, covering a period between 2002 and 2022. An overview of the studies can be found in Table 2. The number of articles and the number of interventions differ, as two articles, that is, Osman et al. (2017, 2021) focus on the same intervention/program. We have included both articles as they describe relevant information on the intervention's mechanisms and results. The majority of the literature was published after 2010 ($n = 20$), with the number of articles peaking in 2015 ($n = 3$), 2017 ($n = 4$), and 2021 ($n = 4$). The selected articles comprised quantitative ($n = 9$), mixed methods (MM; $n = 9$), qualitative studies ($n = 6$), and review studies ($n = 3$; one scoping review and two meta-analyses), as shown in Figure 2. Among the quantitative studies, the majority were intervention studies describing randomised controlled trials (RCT) ($n = 8$), and one observational pre- and post-test design of a pilot study. The MM studies included $n = 2$ (exploratory) randomised trials with an embedded qualitative component using interviews and focus groups to assess participants' experiences with the respective interventions. Two MM studies adopted a longitudinal cohort design combining quantitative with qualitative data. The qualitative studies adopted different study designs (i.e., a case study, interview studies ($n = 3$) and qualitative evaluations of pilot studies ($n = 2$)). The majority of the studies described interventions in a single country, that is, the UK ($n = 13$), the Netherlands ($n = 5$), Sweden ($n = 3$), and Ireland ($n = 1$). Four studies focused on multiple countries or had a global scope including Europe.

In terms of intervention content, we first categorised the interventions inductively based on their main intervention content, independent of setting or delivery modes. This resulted in the following distribution: five articles reported on parenting programs (including pre,- peri- and postnatal programs), nine articles on social change intervention in the wider sense, three on lifestyle intervention (physical activity), two on arts-based interventions, three on self-help interventions (of which two were e-health interventions), and two on health education interventions. Looking at when in the course of prevention these interventions were delivered, we classified eight as primary prevention, eight as secondary, and nine as tertiary prevention (see Table 3).

### Population groups

The selected studies addressed a wide variety of population groups and differed in the terminology used to characterise the study population. Studies mainly from the UK addressed Black and Ethnic Minority (BME) populations (Knifton et al., 2010; Lovell et al., 2014; Rabiee et al., 2015; Van de Venter and Buller, 2015; Lwembe et al., 2017). Others addressed people of African and Caribbean origin (Afuwape et al., 2010; de Freitas and Martin, 2015; Mantovani et al., 2017; Edge et al., 2018). Several studies focused on target groups by nationality, such as Turkish (Reijneveld et al., 2003; Kocken et al., 2008; Christodoulou et al., 2018; Eylem et al., 2021), Moroccan (Kocken et al., 2008), Pakistani (Chaudhry et al., 2009; Gater et al., 2010; Khan et al., 2019), Indian (Jacob et al., 2002), Somali (Osman et al., 2017, 2021), and Iraqi migrants (Siddiqui et al., 2019). One qualitative study focused on forced migrants of different origins in a global perspective, including Germany, Greece and Switzerland (Dubus, 2022). We excluded studies solely focusing on asylum-seekers and refugees, however, this study included participants with undocumented residence status next to refugees and asylum-seekers. Three studies included both ethnic minority populations as well as service providers and institutional stakeholders to assess and triangulate their different perspectives as research participants (de Freitas and Martin, 2015; Lwembe et al., 2017; Dubus, 2022). Two intervention studies were inclusive interventions, targeting socially disadvantaged and underserved populations including ethnic minority populations, yet providing disaggregated results (Lovell et al., 2014; Van De Venter and Buller, 2015). Finally, 6 intervention studies were developed and tested exclusively for ethnic minority women (Jacob et al., 2002; Kocken et al., 2008; Chaudhry et al., 2009; Gater et al., 2010; Hesselink et al., 2012; Khan et al., 2019). The three review studies applied a broad definition of target populations, and included a combination of different ethnic minority populations. One study (Baskin et al., 2021) focuses on UK minority populations, using ethnicity descriptors as defined by the UK 2011 Census from the Office of National Statistics (Office for National Statistics, 2011). Applying the same descriptors in combination with those from the United States Census Bureau (United States Census Bureau, 2020), Arundell et al. (2021) enlarged their focus to "black, ethnic minority, migrant, refugee or asylum seeker communities, and people referred to as 'minorities' or defined as belonging to an identified racial or ethnic 'minority group'" in their global review. Degnan et al. (2018) used the broad definition of ethnic group or subculture, being "a minority culture within a larger dominant culture."

### Types of interventions

Using a public health lens, we describe the identified studies on a prevention continuum, as presented above (see Table 3). Two studies were labelled within multiple categories, for example, an intervention combining evidence-based treatment approaches with mental health promotion at the community level (Eylem et al., 2021; Dubus, 2022).

#### Primary prevention and promotion of well-being

We classified eight studies as primary prevention interventions. Given the economic advantages as well as the legal and human right obligations to keep migrants and ethnic minority populations healthy (Agyemang, 2019), interventions that support them in maintaining good mental health are relevant. The identified interventions were quite diverse in terms of their approaches used, intervention strategies, and target populations. Studies were either based on thorough cultural adaptation of already existing evidence-based interventions, such as a Dutch study reporting on the

**Table 2.** Overview of intervention studies – Alphabetical order

| Authors + publication year | Country | Study design | Population + sample size | Intervention | Mechanisms of change | Results | Limitations |
|---|---|---|---|---|---|---|---|
| Afuwape et al., 2010 | United Kingdom | RCT | $N = 40$ members of BME communities from London Borough of Southwark | A needs-led and community-based package of mental health care, which included brief psychological interventions, advocacy and health education | Advocacy and health education improve depression and reduce anxiety | Participants in the intervention group were significantly less severely depressed compared to control group participants at 3 months (GHQ-28 sub-scale severe depression); the rapid access group had significantly better outcomes than the standard access group for two of the SF-36 scales: mental health $p = 0.04$ and vitality $p = 0.01$; significantly better outcomes for the rapid access group for the Mental Health Component score $p = 0.02$ | Small number of participants; imbalance at baseline between groups; potential interviewer bias |
| Arundell et al., 2021 | Global review | Systematic review and development of conceptual typology | $N = 88$ studies on BME adults experiencing symptoms or diagnosed with mental health conditions or receiving psychological treatment | Review and assessment of the effectiveness of culturally adapted psychological interventions for people from ethnic minority groups + development of a conceptual typology | Not applicable | Adapted interventions had significant better outcomes compared to control conditions (waitlist/no intervention); adapted interventions also had significant better outcomes compared to other active treatments. Benefits were also seen in self-help interventions. Interventions including organisation-specific adaptations were found to be more efficacious than interventions that did not incorporate these types of adaptations | Risk for ecological bias: overlooking important distinctions between cultures, experiences and beliefs |
| Baskin et al., 2021 | United Kingdom | Scoping Review | $N = 7$ studies focusing on ethnic minority populations (adults 18–64 years, no severe mental issues) Excluding studies solely focusing on new migrants and refugees | Review of community-centred interventions focused on improving Public mental health interventions for ethnic minority groups in the UK, excluding clinic interventions | Not applicable | Interventions aimed to address social isolation through building peer-to-peer support and social networks, and to overcome structural barriers in accessing care; interventions were delivered by lay health workers, and facilitated linkages to complementary services. Qualitative data commonly found a reduction in social isolation and stress, and improved mood and self-confidence | Omission of grey literature; homogeneous recommendations cannot be made to a culturally and ethnically heterogeneous population |
| Chaudhry et al., 2009 | United Kingdom | Observational pilot study: pre/post-intervention | $N = 9$ British Pakistani women diagnosed with depression | Social group intervention with weekly group sessions of psychoeducation, | Informal social support, mental and physical health education to reduce depression; | Reduction in depression scores (SRQ) pre-intervention to post-intervention: 15 (SD = 3.08) to 11.7 (SD = 5.95), $p = 0.039$; | Not mentioned |

(*Continued*)

| Authors + publication year | Country | Study design | Population + sample size | Intervention | Mechanisms of change | Results | Limitations |
|---|---|---|---|---|---|---|---|
| | | | | personal grooming, exercise and yoga (10 sessions in total) | facilitation of the development of informal networks to enhance social contact and link the participants with appropriate mental health services | Interviews post-intervention diagnosed 2 participants as no longer depressed (SCAN) Anecdotal feedback from the participants: relationships developed between participants and facilitators and the provision of transport were the most important components of the intervention | |
| Christodoulou et al., 2018 | United Kingdom | Discovery interview method and thematic analyses | *N* = 6 Turkish-speaking self-service users of guided self-help | Guided self-help (GSH) to psychological therapies | Not explicitly mentioned | Results indicate the need for better definitions of guided self-help and the role of psychological well-being practitioners herein; language biases of the intervention; the importance of the relationship between mental health complaints and physical complaints; stigma associated with mental health | Sampling bias: small sample and withdrawal; some participants waited for a re-referral or another psychological intervention |
| de Freitas and Martin, 2015 | The Netherlands | Case study: observation, documentary evidence and interviews | *N* = 20 Cape Verdeans affected by psychosocial problems; *N* = 30 institutional stakeholders | Minority user participation in a community-based mental health advocacy project "Project Apoio," created by a user organisation in Rotterdam to promote Cape Verdean migrants' rights and access to mental healthcare | Inclusion of migrants and ethnic minorities in spaces to give citizens a voice in healthcare governance; Participation Chain Model (PCM) | Getting into participatory spaces did not immediately equate with voicing needs and demands. Participants required assistance in building the confidence necessary to take action, within an environment where they felt encouraged to speak their minds. Individual and collective motivations, mobilisation, and empowering dynamics all play a role in facilitating the involvement of users who are marginalised or stigmatised | No cross-case comparison of participatory spaces engaging different marginalised groups, or healthcare settings |
| Degnan et al., 2018 | Existing Western interventions (i.e., influenced by European culture, including Europe, USA, Canada and Australia) | Systematic review and meta-analysis | 46 papers comprising 43 individual studies with 7,828 participants were included; 31 simple RCTs, 12 cluster RCTs, one block RCT and two non-randomised pilot trials | Review and assessment of the nature and efficacy of culturally-adapted psychosocial interventions for schizophrenia – focus on cultural adaptations in schizophrenia | Not applicable | Significant post-treatment effects in favour of adapted interventions for total symptom severity ($n = 2{,}345$, *g*: $-0.23$, 95% confidence interval (CI) $-0.36$ to $-0.09$), positive ($n = 1{,}152$, *g*: $-0.56$, 95% CI $-0.86$ to $-0.26$), negative ($n = 855$, *g*: $-0.39$, 95% CI $-0.63$ to $-0.15$), and general ($n = 525$, *g*: $-0.75$, CI $-1.21$ to $-0.29$) symptoms | Large variation across studies in the quality of reporting of methodology All papers reported some level of adaptation, but often poorly documented |

| Authors + publication year | Country | Study design | Population + sample size | Intervention | Mechanisms of change | Results | Limitations |
|---|---|---|---|---|---|---|---|
| | | | | | | Nine themes emerged from the data on adaptations: (1) language; (2) concepts and illness models; (3) family; (4) cultural norms and practices; (5) communication; (6) context and delivery (7) content; (8) therapeutic alliance. All studies reported adaptations to language | |
| Dubus, 2022 | Germany, Greece, Iceland, Mexico, Switzerland, and the United States | Qualitative Study: thematic content analysis | N = 73 social workers; N = 34 forced migrants (N = 21 refugees, N = 7 asylum seekers, N = 6 undocumented migrants) | CBT and resilience-enhancing interventions | CBT is expected to be an effective treatment approach for depression and anxiety. Adding resilience interventions could be culturally effective, as they consider the contexts and crises encountered by forced migrants encounter. This should lead to better coping skills and use of the resources they have | Results indicate that most social work interventions consisted of short-term case management centred on initial housing and healthcare. Recipients of CBT appreciated the pragmatic approach of case management and corresponded to concrete needs, but recipients suffering from PTSD and those who did not plan to resettle in the specific host country found it less helpful. Resilience-enhancing interventions were experienced to increase sense of self-control, optimism for the future, and reduce anxious symptoms | Study was unable to determine which practices enhance resilience, thus more research is needed |
| Edge et al., 2018 | United Kingdom | A mixed-methods, feasibility cohort study, incorporating focus groups and an expert consensus conference | N = 31 African-Caribbean people diagnosed with schizophrenia and their families or family support members (FSM) | Culturally adapted Family Intervention (CaFI): an extant Family Intervention (FI) model was culturally adapted with key stakeholders using a literature-derived framework. Ten CaFI sessions were offered to each service user and associated family | Systematically developed, but no overall theory of change for the intervention mentioned | The rating of sessions and the qualitative findings indicated that CaFI was acceptable to service users, families, FSMs and healthcare professionals. Proven feasibility of collecting a range of outcomes to inform future trials. Confirmation of CaFI's acceptability by key stakeholders | Lack of a control group and limited sample size; insufficient power to assess efficacy. Non-generalizability of findings beyond target population |
| Eylem et al., 2021 | United Kingdom, The Netherlands | RCT with qualitative component | N = 18 Turkish migrants with mild to moderate suicidal ideation | Culturally adapted e-mental health intervention, based on an evidence-based e-mental health intervention for the general population | CBT plus mindfulness improves the control of thinking and regulates feelings | Suicidal ideation, depression, and hopelessness scores were improved in both intervention and control group. Participants reported better self-management. While they emphasised the therapeutic benefits, the e-intervention's feasibility was perceived to be | Effects of receiving usual care were not monitored. Stigma on mental health was not assessed |

| Authors + publication year | Country | Study design | Population + sample size | Intervention | Mechanisms of change | Results | Limitations |
|---|---|---|---|---|---|---|---|
| | | | | | | low. Main reasons: not having severe suicidal thoughts and not feeling represented by the culturally adapted intervention | |
| Gater et al., 2010 | United Kingdom | RCT and qualitative study | N = 123 British Pakistani women with diagnosed depression (cluster-randomised in 3 arms) | Intervention group 1: social group intervention; intervention group 2: social group intervention combined with antidepressants; Control group: people taking antidepressants prescribed by GP | The creation of social networks to increase social contacts and activities in a culturally acceptable manner; combined with psychoeducation to increase information on depression | No significant effect on reduction of depression (HRSD score). Differences in social functioning were significant at 3 months FU only (greater in social intervention group and combined group than treatment group only). No statistical significant differences between the groups on the remission of depression (only after 3 months FU but not later on) | Loss of power of statistical models due to RCT; small sample size |
| Hesselink et al., 2012 | The Netherlands | Non-randomised trial | N = 239 ethnic Turkish women (N = 119 in HMHB intervention; N = 120 in control group) | "Happy Mothers, Happy Babies" (HMHB) program: Perinatal education program on smoking, infant care, and psychosocial health; consisting of 8 group classes of 2 h, 2 individual contacts of 2 h each (before delivery), and 2 home visits of 1 h each (after delivery) | Not explicitly mentioned | Participation in HMHB program increased knowledge about smoking, intention to engage in SIDS prevention, and short-term SIDS prevention behaviour. The program had a positive effect on mild depressive symptoms, and among those who had at least six contacts with the program. No intervention effect for smoking during pregnancy, smothering, slapping, and shaking of babies, long-term SIDS prevention behaviour, serious depressive symptoms, and parent–child attachment | No comparable test and control group due to recruitment issues |
| Jacob et al., 2002 | United Kingdom | RCT | N = 70; Asian women with depression in primary care settings (GHQ core >3) | Intervention group: psycho-education in primary care compared with treatment as usual (TAU) (not blinded) | Non-medical explanatory models of illness may result in lower rates of detection of common mental disorders Psychoeducation on depression may positively affect perspectives and outcomes of depression | Women in the intervention group who received the educational material had a higher recovery rate than the control group (defined as GHQ-12 score ≤ 2): OR 3.4 (95% CI: 1.01 to 11.5) at 2 months FU No difference in explanatory models was observed | Conducted within the limits of a busy primary care practice; mechanisms of change unclear |
| Khan et al., 2019 | United Kingdom | Qualitative interviews and | N = 15 British Pakistani mothers that scored high | CBT-based intervention ("the Positive Health | Based on CBT elements ("here and now") and problem–solving | The intervention was acceptable to this group and improvements in depression | Small study sample; non-controlled pre-post feasibility study design; |

(*Continued*)

**Table 2.** (*Continued*)

| Authors + publication year | Country | Study design | Population + sample size | Intervention | Mechanisms of change | Results | Limitations |
|---|---|---|---|---|---|---|---|
| | | evaluation design | on the EPDS depression scale (>11) | Program" (PHP)) for depression | approach; culturally based therapies for depression | and health-related quality of life were noted. "Depression" was understood in physical terms, triggered by psychosocial causes, such as marital disharmony, lack of social support, and financial difficulties. Antidepressants were offered in the past and not welcomed. The need for culturally sensitive interventions and limited cultural sensitivity of NHS staff was stressed | measurements of assessment tools; no data collection on antidepressants/nature of depression |
| Knifton et al., 2010 | Scotland | Evaluation design: Mixed methods (focus group discussions and attitude scales) | N = 257; members of the major black and minority ethnic (BME) communities, that migrated from India, Pakistan and China | Intervention group: 26 mental health awareness workshops organised by community members | Active involvement of members of BME communities in the organisation and awareness raising of mental health decreases mental health stigma | "Community conversation" workshops effectively engaged participants and resulted in reductions in reported stigma | Sample biases: in terms of gender, generation, participation/ marginalisation in communities; no population-wide awareness campaign |
| Kocken et al., 2008 | The Netherlands | RCT + (provider-administered interviews in patients' own language) | N = 104 female patients with psychosomatic disorders from Turkish and Moroccan origin | Intervention group: 8 group sessions by trained migrant health educators, and individual tailored counselling including a conclusion and evaluation session Control group: TAU | Based on stress reduction theory migrant health educators were used to improve communication with GPs, change beliefs (explanatory models), self-efficacy to cope with stressors and coping with psychosomatic complaints | Significant improvement of perceived general and psychological health and reported ability to cope with pain were observed in the intervention group. No effects for social support and the perceived burden of stressful life-events | Study design does not allow to determine which intervention elements caused the effects. Short follow-up period |
| Lovell et al., 2014 | United Kingdom | Exploratory randomised trial with mixed-methods analysis | N = 57 participants; n = 20 belonging to ethnic minorities | An acceptable and culturally sensitive psychosocial intervention for older people and people from ethnic minority communities. Participants were offered an initial patient-centred assessment session with a well-being facilitator, and collaboratively devised a well-being plan | Culturally adapted CBT elements; patients as agents of change | Effects among ethnic minorities were generally smaller than in the elderlies. The largest effects were on depression (PHQ9), health-related quality of life (EQ5D) and functioning (WSAS) Qualitative analysis results: importance of face-to-face contact and flexibility to consult people at home; group intervention and content of intervention positively evaluated, need for longer-term support | Scales not culturally validated; low recruitment rates due to constraints in resources/limited time |

(*Continued*)

*Cambridge Prisms: Global Mental Health*

| Authors + publication year | Country | Study design | Population + sample size | Intervention | Mechanisms of change | Results | Limitations |
|---|---|---|---|---|---|---|---|
| Lwembe et al., 2017 | United Kingdom | Pilot study evaluation (focus group discussions and semi-structured interviews) | $N = 25$ (Interviews with patients of BME, community group representatives, providers ($N = 19$); and one focus group with $N = 6$ patients | Participation in the co-production of a novel community mental health service for black and ethnic minority service users | Developing personal goals and participation in decisions to increase sense of control to enable patients to access services and complete treatment | Co-production of all stakeholders involved better responds to the needs of BME and reduces mental health inequalities and service use. Co-production can be integral to mental health service delivery | Small-scale study with limited intervention period |
| Malone et al., 2017 | Ireland | A combined and integrated science-arts study design | $N = 150$ members of the Irish Travellers community | Combination of a psycho-biographical autopsy with a visual arts autopsy, in which families donated stories, images and objects associated with the lived life of a loved one lost to suicide. Through an interdisciplinary research platform, a mediated exhibition was created (Lived Lives) around suicide prevention | Not explicitly mentioned | The intervention demonstrated that hard-to-reach audiences can be reached and engaged on sensitive health issues such as suicide. The Lived Lives methodology encouraged inclusivity from the start and the Irish Travellers involved took co-ownership of the project and moved towards taking active steps to address the problem of suicide in their community | Practical drawbacks to the location and timing of the art event |
| Mantovani et al., 2017 | United Kingdom | Qualitative study using a participatory approach | $N = 13$ African and African-Caribbean lay people | Pilot outreach intervention: Community engagement model to address mental health needs Lay people trained as well-being champions to raise awareness about mental health in the community; Methods used to achieve buy-in in faith-based organisations: workshops, awareness raising, and meetings | Logic model of change is mentioned, but not explained in detail: nonlinear, reciprocal relationship between community engagement processes and the social determinants of mental health | Community champions used group work and informal one-to-one conversations as main strategies. Circles of influence were used to share ideas about mental health and well-being. Community champions encountered resistance at community level: lack of knowledge on mental health, taboos and ascribed stigma. Community champions felt inadequately equipped to address sensitive issues. They were instrumental in bringing people together, formed a network structure and some acted as a bridge to public health services | No longitudinal evaluation of the intervention and lack of insight in transformative changes within community engagement |
| Osman et al., 2017 | Sweden | RCT | $N = 120$ Somali-born parents with children school-aged children (11–16 yrs) with self-perceived parenting stress ($N = 60$ in intervention group; and | An intervention with two main components: societal information and the existing CONNECT parenting program, delivered using a culturally | Attachment theory, parents are encouraged to focus on strengthening the parent–child relationship | Significant mental health improvements in the intervention group compared with control parents at 2-month follow-up: $B = 3.62$, 95% CI 2.01 to 5.18, $p < 0.001$. Significant improvement was | Short interval of follow-up (2 months) |

**Table 2.** (*Continued*)

| Authors + publication year | Country | Study design | Population + sample size | Intervention | Mechanisms of change | Results | Limitations |
|---|---|---|---|---|---|---|---|
| | | | $N = 60$ in waiting list control group) | sensitive approach (12 group-based sessions). The intervention was culturally adapted (eg. Somali facilitators of both sexes) and gender-mixed groups. Childcare services were offered during sessions | | found for efficacy ($B = -6.72$, 95%CI $-8.15$ to $-5.28$, $p < 0.001$) and satisfaction ($B = -4.48$, 95%CI $-6.27$ to $-2.69$, $p < 0.001$) compared to controls. Parents' satisfaction mediated the intervention effect on parental mental health ($\beta = -0.88$, 95%CI $-1.84$ to $-0.16$, $p = 0.047$). No gender differences between mothers and fathers | |
| Osman et al., 2021 | Sweden | Longitudinal cohort study design (Impact study of Osman et al., 2017) | $N = 51$ Somali-born parents (participants in study above: Osman et al., 2017) | (See above, Osman et al., 2017) | (See above, Osman et al., 2017) | Significant improvement over time for all outcome including parents' mental health (GHQ-12: 95%CI 0.40 to 3.11, $d = 0.46$), and their children were maintained 3 years after the intervention. | Lack of control group; outcome measures not previously tested for reliability and validity on specific population; use of parental reports only may introduce a bias |
| Rabiee et al., 2015 | United Kingdom | Cross-sectional mixed methods design | $N = 257$ members from ethnic minority groups in a deprived constituency in Birmingham | Gym-for-free pilot project providing adults free access to leisure centres | Regular exercise helps to improve mood, self-esteem, confidence and quality of life | Increased energy levels, confidence, mental well-being, reduction in stress and anxiety, improved stress relief and anger management were reported. The pilot scheme increased the uptake of exercise particularly for women in an economically deprived inner city area, especially those from Pakistani and Bangladeshi ethnic backgrounds. The use of leisure facilities also increased markedly ($p < .05$) | Long-term evaluation is required; recruitment was opportunistic, not random and not generalizable |
| Reijneveld et al., 2003 | The Netherlands | RCT | $N = 92$; Turkish first-generation elderly immigrants (aged 45+) | Intervention group: culturally adapted evidence-based program compared with a non-adapted physical exercise program | Not explicitly mentioned | Improvement in mental health (effect size: 0.38 SD (95% confidence intervals 0.03 to 0.73), $p = 0.03$) in the intervention group; in the oldest subgroup also in mental well-being (effect size 0.75 SD (0.22 to 1.28), $p = 0.01$). Effects were largest for participants aged above 55 | Measurements for physical activity were modified, which may explain the negative outcomes (no improvements observed) |
| Siddiqui et al., 2019 | Sweden | RCT | $N = 96$ Iraqi immigrants with a high risk for diabetes and depression (Intervention $N = 50$; control group, $N = 46$). | Seven group sessions addressing self-empowerment with emphasis on social interaction, social support and | Increased release of neurotransmitters through physical activity to improve mental health, and psychological | Intervention group significantly scored lower on MADRS-S and HADS depression scale at visit 3 (MADRS-S OR 5.9, 95% CI: 1.6–22.5; HADS OR 4.4, 95% CI: 0.9– | Study not designed for severe depression and anxiety; short follow-up duration (4 months, effects at 2 months were not significant) |

(*Continued*)

**Table 2.** (*Continued*)

| Authors + publication year | Country | Study design | Population + sample size | Intervention | Mechanisms of change | Results | Limitations |
|---|---|---|---|---|---|---|---|
| | | | | motivation, conducted over a period of 4 months with intervals of 1–4 weeks. The control group received written information | mechanisms: social interaction and social support to improve self-esteem and self-efficacy | 20.3). The results persisted after adjustment for age, sex, body mass index, time since migration, sedentary lifestyle and language spoken at home | |
| Ünlü Ince et al., 2013 | The Netherlands | RCT | *N* = 96 Turkish adults with depressive symptoms (*N* = 49 experimental group and *N* = 47 control group) | Intervention group: a culturally sensitive guided self-help, problem-solving intervention through the internet for reducing depressive symptoms in Turkish migrants; control group: waiting list | CBT and problem-solving therapy; culturally adapted | No statistically significant effects on the reduction of depressive symptoms, but the effect size at the post-test was high (indicator of effectiveness if tested with sufficient sample size?) | Small sample size |
| Van De Venter and Buller, 2015 | United Kingdom | Pre- and post-intervention mixed-methods pilot study | *N* = 44 (White (*N* = 38) and BME (*N* = 6) British participants with mild/moderate mental health symptoms | Intervention: arts-on-referral (AoR) scheme | Not explicitly mentioned: Art interventions were found to improve mental well-being in a cost-effective way | Participation in AoR improved mean well-being and in Warwick Edinburgh Mental Well-being Scale (WEMWBS) scores per session, however, more slowly for those with low baseline scores. Nonetheless, the latter may find arts participation helpful in managing emotions and preventing deterioration of well-being | Small sample size (especially for qualitative research, *N* = 6), lack of control group, election bias and the lack of a power calculation |

Abbreviations: AoR, arts-on-referral; BME, black and minority ethnic, British ethnic minority population; CaFI, culturally adapted family intervention; CBT, cognitive behavioural therapy; CI, confidence interval; EPDS, Edinburgh postnatal depression scale; EQ-5D, European quality of life – 5 dimensions; FI, family intervention; FSM, family support members; FU, follow-up; GHQ-28/12, general health questionnaire (28/12 items); GP, general practitioner; GSH, guided self-help; HADS, hospital anxiety and depression scale; HMHB, "happy mothers, happy babies"-programme; HRSD, Hamilton rating scale for depression; MADRS-S, Montgomery–Asberg depression rating scale; NHS, National Health Service; OR, odds ratio; PHP, positive health program; PCM, participation chain model; PHQ-9, patient health questionnaire; PTSD, post-traumatic stress disorder; RCT, randomised controlled trial; SCAN, schedule for clinical assessment in neuropsychiatry; SD, standard deviation; SF-36, short form survey (36 items); SIDS, sudden infant death syndrome; SRQ, self reporting questionnaire; TAU, treatment as usual; WEMWBS, Warwick Edinburgh mental well-being scale; WSAS, work and social adjustment scale.

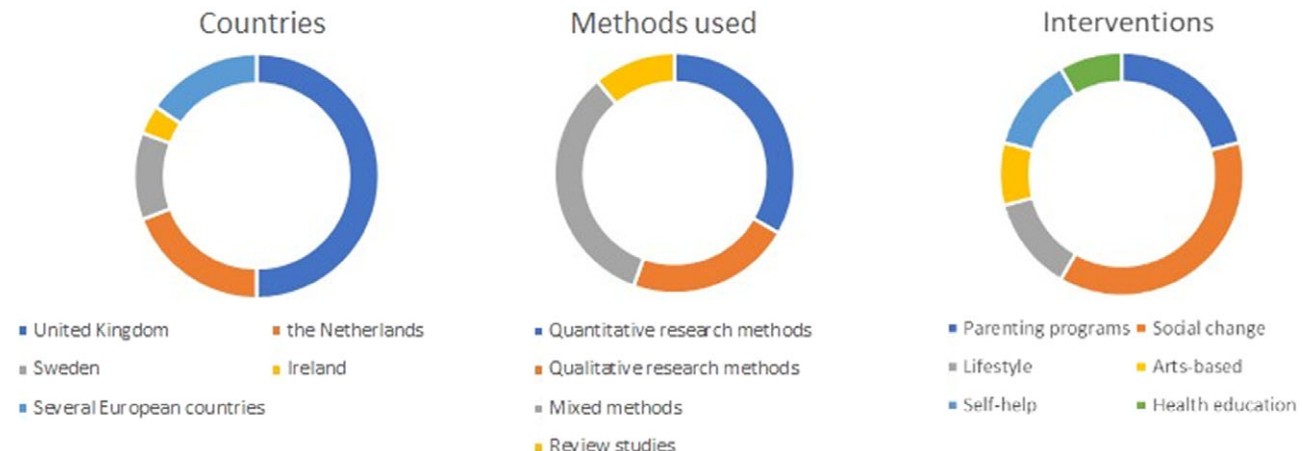

**Figure 2.** Countries, methods used and intervention content of the included studies.

**Table 3.** Categorisation of studies

| Authors + publication year | Method | | | Prevention | | | Participatory approach | Cultural adaptation of interventions |
| | Qualitative | Quantitative | Mixed Methods | Primary | Secondary | Tertiary | | |
|---|---|---|---|---|---|---|---|---|
| Afuwape et al., 2010 | | X | | | | X | CI | |
| Arundell et al., 2021 (review) | | X | | | | X | CI/CS/CD | X |
| Baskin et al., 2021 (review) | X | | | X | X | X | CS | |
| Chaudhry et al., 2009 | | X | | | | X | CS | |
| Christodoulou et al., 2018 | X | | | X | | | CI | |
| de Freitas and Martin, 2015 | X | | | | X | | CD | |
| Degnan et al., 2018 (review) | | X | | | | X | CI/CS/CD | X |
| Dubus, 2022 | X | | | | X | X | | |
| Edge et al., 2018 | | | X | | | X | CD | X |
| Eylem et al., 2021 | | | X | | X | X | | X |
| Gater et al., 2010 | | | X | | | X | CI | |
| Hesselink et al., 2012 | | X | | X | | | | |
| Jacob et al., 2002 | | X | | | | X | | |
| Khan et al., 2019 | | | X | | | X | | X |
| Knifton et al., 2010 | | | X | X | | | CD | |
| Kocken et al., 2008 | | X | | | X | | | |
| Lovell et al., 2014 | | | X | X | | | CI | X |
| Lwembe et al., 2017 | X | | | | X | | CI | |
| Malone et al., 2017 | X | | | X | | | CD | |
| Mantovani et al., 2017 | X | | | X | | | CD | X |
| Osman et al., 2017 | | X | | | X | | | X |
| Osman et al., 2021 | | | X | | X | | | |
| Rabiee et al., 2015 | | | X | X | | | | |
| Reijneveld et al., 2003 | | X | | X | | | | X |
| Siddiqui et al., 2019 | | X | | | X | | | X |
| Ünlü Ince et al., 2013 | | X | | | | X | CI | X |
| Van De Venter and Buller, 2015 | | | X | | X | | | |

Abbreviations: CD, community-driven; CI, community-informed; CS, community-shaped.

successful cultural adaptation of a physical exercise program for elderly Turkish migrants in the Netherlands (Reijneveld et al., 2003). The intervention resulted in significant mental health improvements, but not in physical well-being and exercise activity (see Table 2). An exploratory primary care trial tested a well-being intervention in general practitioners (GP)-settings in the UK (Lovell et al., 2014) including both underserved older Europeans and people of Somali- or South Asian origin. This intervention combined individual- and group elements with adequate referral and resulted in improved well-being and social functioning in both groups. However, community engagement turned out to be a more relevant factor in the migrant group than in the comparison group. Three studies described community-led interventions emphasising participatory approaches (Knifton et al., 2010; Malone et al., 2017; Mantovani et al., 2017). The latter used arts-based strategies to engage communities, while Mantovani et al. (2017) adopted a community engagement model to train and work with well-being champions. Earlier, Knifton et al. (2010) engaged community members to hold awareness-raising workshops effectively addressing mental health stigma and discrimination. A culturally sensitive perinatal program systematically developed for pregnant Turkish mothers ("Healthy mothers, healthy babies") engaged ethnic minority midwives to conduct group sessions and home visits (Hesselink et al., 2012). The intervention delivered in mother–child centres demonstrated a positive effect for mild depressive symptoms, but was underpowered to detect differences in other envisaged outcomes (i.e., severe depression, parenting behaviours, smoking cessation).

## Secondary prevention

Eight articles qualified as targeting people and groups who are at heightened risk of developing mental health problems. The mental health and psychosocial problems addressed differed according to studies and targeted populations. Conditions and population groups addressed included PTSD among forced migrants of diverse origins in a multi-country study using a resilience approach (Dubus, 2022), acculturation stress among Somali-born parents in Sweden (Osman et al., 2017, 2021), concurrent diabetes and depression among Iraqi immigrants in Sweden (Siddiqui et al., 2019), suicidal ideation among Turkish migrants in the Netherlands (Eylem et al., 2021), and psychosomatic problems and pain among Turkish and Moroccan women in the Netherlands (Kocken et al., 2008).

One study looked at access to mental health services for people diagnosed with mental illness (Lwembe et al., 2017) using co-production techniques. A qualitative study explored how Cape Verdean migrants experiencing psychosocial problems (de Freitas and Martin, 2015) could be encouraged, valued, and sustained through participatory initiatives by creating community-based hybrid mental health spaces. A culturally adapted health education intervention delivered in primary care settings in the Netherlands used trained migrant educators providing culturally adapted information, counselling and support based on stress reduction theory for women of Turkish and Moroccan origin with psychosomatic problems (Kocken et al., 2008). The randomised controlled trial showed significant improvements in perceived general and psychological health, and self-reported ability to cope with pain in the intervention group compared to a control group receiving treatment as usual (TAU). No effects were found on social support and the perceived burden of stressful life events due to precarious life circumstances. The evaluation revealed participants' subjective perceptions of psychosomatic problems due to their different explanatory mental health models, as well as difficulties to change social support from the women's environment due to their often difficult socio-economic situation.

## Tertiary prevention and self-management

We identified nine articles focusing primarily on tertiary prevention. Since the continuum is fluid, two studies (Eylem et al., 2021; Dubus, 2022) covered both secondary and tertiary prevention. Importantly, these studies showed how the use of migrant community health workers (e.g., Jacob et al., 2002; Afuwape et al., 2010; Gater et al., 2010) can help to target social determinants underlying mental health conditions. Studies were targeting families and social groups: two studies described family interventions: one study addressed women with postnatal maternal depression using cognitive-behavioural therapy (Khan et al., 2019), and Edge et al. (2018) reported on a systematically developed family-based intervention for schizophrenia. Two studies reported on social group interventions (Chaudhry et al., 2009; Gater et al., 2010), the latter in a primary care setting. Two interventions trained and employed community health workers (Afuwape et al., 2010), or trained migrant health educators (Jacob et al., 2002). Finally, we identified two culturally adapted self-management interventions using online technologies. The online intervention by Eylem et al. (2021) used culturally adapted elements of an already existing evidence-based e-intervention to reduce suicidal ideation coupling Cognitive Behavioural Therapy (CBT) with mindfulness practices. The intervention was evaluated using an RCT design with waiting-list control condition both in the UK and the Netherlands targeting Turkish migrants at risk of suicide. It showed improved suicidal ideation, depression, and hopelessness scores in both groups, no suicide attempt was reported during the study period, and participants reported better self-management. Participants perceived the mindfulness practices as helpful but reported that the online intervention provided too little structure while not being diversified enough. This pointed to the heterogeneity of migrant communities, and the existence of specific micro-identities. Another UK-based study (Afuwape et al., 2010) tested the feasibility and effectiveness of a culturally acceptable package of mental health care to improve the health and psychosocial functioning among BME patients, mainly of sub-Saharan African descent with a previous history of diagnosed depression and/or anxiety. Trained community health workers (i.e., ethnically matched psychology graduates) delivered brief CBT interventions under supervision), as well as advocacy and mentoring creating rapid access. This small-scale randomised community trial comparing an intervention group with TAU (i.e., local mental health services) showed significantly improved levels of depression at the 3 months follow-up for the rapid access group. It was the only study including a cost-effectiveness component, demonstrating that a needs-led mental health package did not significantly increase costs of service use. This intervention also improved the interface between statutory agencies and African community organisations, which is relevant from a policy point of view.

## Intervention mechanisms: Possible pathways to effective intervention outcomes

### Interventions' effectiveness

Three reviews looked at the effectiveness of interventions (Degnan et al., 2018; Arundell et al., 2021; Baskin et al., 2021). Two meta-analyses demonstrated significant improvements of culturally adapted interventions over time compared to non-adapted interventions: one

systematic review including 46 studies with more than 7,800 participants looked at post-treatment effects of culturally adapted psychosocial interventions for patients living with schizophrenia. The review showed significant post-treatment improvements for total symptom severity over interventions that were not explicitly mentioned to be adapted for the specific cultural population groups (Degnan et al., 2018). A second systematic review (Arundell et al., 2021) synthesised 88 studies describing psychological interventions for people belonging to ethnic minority populations experiencing or being diagnosed with a wide range of mental health problems (depression, anxiety, post-traumatic stress syndrome, psychosis, eating disorders and other non-specified mental health problems) on a global scale. The meta-analysis found medium effect size in reducing symptom severity in favour of adapted interventions when compared to controls, across all target conditions and adaptation types including self-help interventions. A narrative scoping review (Baskin et al., 2021) looked at the effectiveness of interventions using a community-centred approach in the UK. The authors identified seven studies, including four studies reporting statistically significant positive effects on mental health outcomes. Social connectedness, access to safe and affordable housing, and power in local decision-making were reported as important determinants for intervention effectiveness.

Among the studies identified in our review, several studies also showed significant improvements in mental health: six of the seven intervention studies adopting an RCT design demonstrated a positive effect on mental health outcomes, mostly a reduction in depression rates (Jacob et al., 2002; Reijneveld et al., 2003; Kocken et al., 2008; Afuwape et al., 2010; Osman et al., 2017; Siddiqui et al., 2019). Also, a non-randomised trial study (Hesselink et al., 2012) showed positive effects on reduction of mild depressive symptoms.

This points to an increasing evidence-base of effective interventions, at least under the controlled circumstances of trial studies. Interestingly, the four lifestyle interventions focusing on regular physical activity to also improve mental health outcomes, were all able to demonstrate positive impact on mental health outcomes (Reijneveld et al., 2003; Osman et al., 2017, 2021; Siddiqui et al., 2019). Also, the pilot evaluation of a free access scheme to exercise facilities for BME communities in the UK showed a preliminary increase in energy levels and self-confidence as well as reduction in stress, depression and anxiety (Rabiee et al., 2015). These findings show that social and environmental circumstances are integral to lifestyle choices, hence the importance of public health policy to facilitate the joining up of different organisations to increase access and offer tailored activities.

Different factors explained why some interventions did not produce significant improvements on mental health outcomes, either referring to flaws in the study design or in the difficulty to impact structural and social factors. Some studies reported improvements in both intervention and control groups (Ünlü Ince et al., 2013; Eylem et al., 2021), showing the difficulty of conducting an RCT in real-life circumstances as community and social influences cannot be excluded (Ünlü Ince et al., 2013). Studies did not assess the exposure to usual care, provide sufficient cultural adaptation or assess mental health stigma (Eylem et al., 2021), or had problems recruiting sufficient participants (Hesselink et al., 2012). While direct support from migrant health educators contributed to the improved effects on coping and mental health, social support from participants' direct personal environment as one of the determinants of mental health problems proved to be more difficult to be influenced (Kocken et al., 2008).

### Theory-driven interventions and intervention mechanisms

We analysed the theory-driven processes underlying the interventions, and the hypothesised processes leading to the observed outcomes. Evidence shows that complex healthcare interventions are more likely to be effective, sustainable, and scalable if they are using a sound theory-base and describe and test the causal pathways through which an intervention may achieve its expected outcome (De Silva et al., 2014). Providing such information also makes the intervention replicable, increasing knowledge on both the interventions' mechanisms and practical implementation. Against this background, we took a rather broad approach and coded whether studies provided any information on their underlying theory-base for assumed causal changes achieved through the intervention. Evidence also shows that developing, implementing, and evaluating interventions in collaboration with stakeholders adds to their effectiveness (Bartholomew et al., 2016). Thus, we also coded stakeholder participation (i.e., any pragmatic framework or narrative description explaining how the intervention may affect change).

More than half ($n = 16$) of the selected articles on intervention studies explicitly described their underlying theory base and hypothesised intervention mechanisms. Parenting interventions for instance, were based on attachment theory to support distressed parents in their adaptation to parenting styles in the host country change (i.e., Sweden) (Osman et al., 2017). For lifestyle interventions working with physical exercise, a combination of neuro-biological, psychological and social mechanisms was mentioned: enhanced physical activity leads to increased release of neurotransmitters believed to improve mental health, and psychological mechanisms such as social interaction and social support to improve self-esteem and self-efficacy (Rabiee et al., 2015; Osman et al., 2017). This resulted in self-empowerment, which indeed showed improved scores in depression outcomes among Iraqi immigrants from baseline to the 3 months follow-up (Siddiqui et al., 2019). Some studies also focused on the improvement of communication patterns between mental health care providers and patients due to culturally diverse explanatory models, mainly through cultural mediators, community health workers and well-being champions (Jacob et al., 2002; Kocken et al., 2008; Mantovani et al., 2017).

Several interventions aimed at improving mental health outcomes through the creation of social networks to increase social contacts and activities in a culturally acceptable manner, therefore reducing social isolation. Combining such elements with psychoeducation to increase correct information on depression resulted in significant reduction of depression in a social intervention delivered in primary care settings for British Pakistani women with depression at 3 and 9 months follow-up (Gater et al., 2010). Chaudhry et al. (2009) mentioned a similar theory of change: providing mental and physical health education and facilitating the development of informal networks to increase engagement in social contacts would reduce depression; they also linked the participants to appropriate mental health services to increase access. This intervention showed a significant reduction in depression scores from pre-to post-test, and feedback from the nine British Pakistani women with diagnosed depressive disorders showed that they perceived their relationship with the group session facilitators and the provision of transport as the most important components of the intervention. Some studies did not explicitly describe their underlying change models, but mentioned that they were systematically developed, or that they had conducted their own need assessment (Hesselink et al., 2012; Edge et al., 2018) or a cyclic

process of data collection and evaluation (Christodoulou et al., 2018). Specifically, Kocken et al. (2008) recommended using generic guiding frameworks for the systematic development of health promotion interventions, such as the intervention mapping protocol to effectively tailor interventions to migrants' needs.

The underlying theory-base specifying the respective determinants that interventions aim to address to achieve the envisaged behavioural outcomes is also relevant as it determines the choice of the respective intervention strategies (Bartholomew et al., 2016): $n = 7$ intervention studies explicitly mentioned using cognitive behavioural therapy (CBT) approaches (Afuwape et al., 2010; Ünlü Ince et al., 2013; Lovell et al., 2014; Khan et al., 2019; Eylem et al., 2021; Dubus, 2022) or strategies that used CBT elements, such as personalised goal setting. As there is a large body of evidence for the effectiveness of CBT in treating mental health conditions, some of these interventions were based on existing evidence-based interventions, which were culturally adapted (Ünlü Ince et al., 2013; Khan et al., 2019; Eylem et al., 2021), other studies developed new interventions using participatory approaches (Afuwape et al., 2010; Dubus, 2022). The second main strategy consisted of various peer-support and participatory strategies ($n = 6$) focusing on empowerment through the facilitation of social interaction and social support (Gater et al., 2010; Knifton et al., 2010; de Freitas and Martin, 2015; Lwembe et al., 2017; Mantovani et al., 2017; Siddiqui et al., 2019).

Table 4 summarises and ranks the different interventions and their change mechanisms (if mentioned in the studies) according to the level of social determinants they addressed: from the micro-level addressing individual level-factors, such as health education to change lifestyles, to interventions on the meso-level, focusing on migrants' and ethnic minorities' social and community networks, to interventions on a macro-level focusing on health systems changes through, for example, community participation and shared decision-making. We have adapted these levels from the often-applied social determinant framework, coined by Dahlgren and Whitehead (1991). Some interventions addressed multiple levels of social determinants, for example, Gater et al. (2010) created social networks and increased the participants' knowledge on depression. For conceptual clarity, we distinguish between these levels, yet clearly they interact with each other and ultimately influence mental health at the individual level (Glanz et al., 2008).

### Cultural adaptation of interventions

In general, we could distinguish two types of studies: interventions based on a cultural adaptation of already existing evidence-based interventions and newly developed interventions specifically designed for a certain population group. Those studies adapting existing interventions in a culturally sensitive way did so to various degrees: ranging from the design of the intervention, as a feedback process, during the implementation, adjusting language and translation issues or dealing with the (socio-economic) preconditions to enable participation in the intervention. In addition, interventions were frequently adjusted to a specific culture or target group, taking into account local habits, languages and explanatory models.

Eight studies discussed the cultural adaptation of an existing intervention, see Table 3. As mentioned above, CBT approaches were modified for the specific target groups. Importantly, adopting CBT elements was independent of who delivered the intervention (e.g., expert patients as co-facilitators or professionals such as therapists or GPs) or in which type of setting, thus including e-health interventions (Ünlü Ince et al., 2013; Eylem et al., 2021),

**Table 4.** Interventions and their underlying intervention mechanisms ranked according to different levels of social determinants

| Interventions | Underlying intervention mechanisms |
|---|---|
| **Individual lifestyle** | |
| Stimulating physical exercise (Chaudhry et al., 2009; Rabiee et al., 2015; Osman et al., 2017; Siddiqui et al., 2019) | Physical activity is suggested to enhance release of beneficial neurotransmitters, improves mood, self-esteem, and confidence |
| Stimulating art activities (Van de Venter and Buller, 2015; Malone et al., 2017) | Art interventions reach and engage hard-to-reach audiences, helps to regulate emotions, and improve well-being |
| Improving health(care) knowledge (Jacob et al., 2002; Chaudhry et al., 2009; Afuwape et al., 2010; Gater et al., 2010; Hesselink et al., 2012) | Psychoeducation enhances health(care) knowledge |
| **Social and community networks and engagement** | |
| Creation of social networks (Chaudhry et al., 2009; Gater et al., 2010; Siddiqui et al., 2019) | Increased social contacts reduce social isolation, improve self-esteem and self-efficacy |
| Improving parenting skills (Osman et al., 2017, 2021) | Adapting parenting skills (attachment theory) improves parent–child relation |
| Community engagement (Knifton et al., 2010; Malone et al., 2017; Mantovani et al., 2017) | Community engagement increases awareness, improves communication and linkage with healthcare, changes beliefs, improves self-efficacy in coping |
| Addressing mental health stigma in communities and society (Knifton et al., 2010) | Awareness raising and addressing diverse cultural understandings of mental health reduces stigma |
| **Adjusting the health system to the needs of migrants and ethnic minorities** | |
| Improving communication between healthcare providers and migrants and ethnic minorities (Jacob et al., 2002; Kocken et al., 2008; Mantovani et al., 2017) | Cultural mediators, community health workers, and well-being champignons improve the communication and link to healthcare services, by taking cultural explanatory models of health into account |
| Linking people to appropriate healthcare services, improving access to healthcare (Chaudhry et al., 2009; Lovell et al., 2014; de Freitas and Martin, 2015; Christodoulou et al., 2018) | Improved access and linkage in services improves uptake of services |
| (Adapting) effective mental healthcare therapies (Reijneveld et al., 2003; Kocken et al., 2008; Afuwape et al., 2010; Ünlü Ince et al., 2013; Lovell et al., 2014; Osman et al., 2017; Khan et al., 2019; Eylem et al., 2021; Dubus, 2022) | Cultural adaptation of existing services/culturally developed services improve communication, coping skills, uptake, and effectiveness of services |
| Inclusion of migrants and ethnic minorities in (mental) healthcare governance (de Freitas and Martin, 2015; Edge et al., 2018; Lwembe et al., 2017) | Participation in governance and decision-making increases sense of control, access to services and service uptake |

a CBT-based therapy (Khan et al., 2019) or a well-being program for primary care (Lovell et al., 2014). Osman et al. (2017, 2021) based their intervention on the evidence-based parenting programme CONNECT, which they delivered using a culturally

sensitive approach. In the study of Edge et al. (2018), an existing family intervention was adapted by applying a participatory approach with African-Caribbean people diagnosed with schizophrenia, their families, service providers and researchers. Lifestyle interventions were adapted by Siddiqui et al. (2019), focusing on healthy lifestyle habits, and Reijneveld et al. (2003), who adapted a physical exercise program called "Healthy and Vital program." In addition to these single studies, also two review studies discuss culturally adapted interventions (Degnan et al., 2018; Arundell et al., 2021). Degnan et al. (2018) focus on adapted psychosocial interventions for schizophrenia, assessing their effectiveness (see above) and proposing a framework for cultural adaptation. Arundell et al. (2021) determined the effectiveness of cultural adaptations in psychological intervention for BME groups (see above) and developed a conceptual typology.

The strategies to culturally adapt interventions varied over the different phases of intervention studies, see Table 5. In a pre-development stage, preparatory focus group discussions with community members were held to identify adaptation needs (Reijneveld et al., 2003; Lovell et al., 2014; Edge et al., 2018). In the process of intervention development, adaptations made included the (back)

**Table 5.** Strategies for cultural adaptation of existing evidence-based interventions

| Strategies for cultural adaptation | |
| --- | --- |
| **Pre-development** | |
| Preparatory focus group discussions with community members | Reijneveld et al., 2003; Lovell et al., 2014; Edge et al., 2018 |
| **During development** | |
| Back-translation of intervention materials | Ünlü Ince et al., 2013; Siddiqui et al., 2019; Eylem et al., 2021 |
| Modifications of content<br>• modifying concepts and including well-known idioms and metaphors<br>• incorporating culture-specific norms and practices<br>• including cultural models of mental health and illness<br>• incorporating religious or spiritual beliefs | Ünlü Ince et al., 2013; Lovell et al., 2014; Degnan et al., 2018; Edge et al., 2018; Khan et al., 2019; Arundell et al., 2021; Eylem et al., 2021 |
| **During intervention implementation** | |
| Adaptation of communication strategies<br>• Use of culturally sensitive language | Ünlü Ince et al., 2013; Degnan et al., 2018; Edge et al., 2018; Khan et al., 2019 |
| Educational approach to close socio-cultural knowledge gaps | Reijneveld et al., 2003; Osman et al., 2017; Edge et al., 2018; Siddiqui et al., 2019; |
| Establishing of culture-appropriate alliance<br>• Involvement of professionals with similar background or language<br>• Involvement of culturally competent professionals | Degnan et al., 2018; Arundell et al., 2021 |
| Practical adaptations<br>• Accessible locations<br>• Adjustment of length/timing intervention<br>• Economic support | Degnan et al., 2018; Edge et al., 2018; Siddiqui et al., 2019; Arundell et al., 2021 |

translation of intervention materials (Ünlü Ince et al., 2013; Siddiqui et al., 2019; Eylem et al., 2021), modifications in content (Degnan et al., 2018; Arundell et al., 2021) such as modifying concepts and including well-known idioms and metaphors (Ünlü Ince et al., 2013; Lovell et al., 2014; Khan et al., 2019; Eylem et al., 2021), incorporating culture-specific norms and practices (Degnan et al., 2018; Edge et al., 2018;), including cultural models of mental health and illness (Degnan et al., 2018; Edge et al., 2018) and incorporating a broader perspective by including religious or spiritual beliefs (Degnan et al., 2018; Edge et al., 2018; Arundell et al., 2021). During the implementation of interventions, communication strategies were adapted (Degnan et al., 2018; Edge et al., 2018) with attention to culturally-sensitive language use (Ünlü Ince et al., 2013; Degnan et al., 2018; Edge et al., 2018; Khan et al., 2019), socio-cultural barriers and knowledge gaps were addressed through educational approaches (Reijneveld et al., 2003; Osman et al., 2017; Edge et al., 2018; Siddiqui et al., 2019), and investments were made in establishing a culture-appropriate (therapeutic) alliance (Degnan et al., 2018; Arundell et al., 2021). This was done through the involvement of professionals with either a similar background or the same native language knowledge as the target groups (Reijneveld et al., 2003; Osman et al., 2017; Siddiqui et al., 2019) or professionals being culturally competent or "at least 'culturally aware'" (Edge et al., 2018). Attention was paid to cultural sensitiveness in therapeutic assignments, examples, and case stories (Ünlü Ince et al., 2013; Osman et al., 2017; Khan et al., 2019). Some studies adopted a holistic approach to intervention delivery, involving families or a broader social network (Degnan et al., 2018; Edge et al., 2018). Also, (practical) adaptations were made to increase the intervention's feasibility, such as assuring accessible locations (Degnan et al., 2018; Arundell et al., 2021), adjusting the length or timing of the intervention (Arundell et al., 2021) or providing economic support (Siddiqui et al., 2019).

### Participatory approaches

A total of 15 studies described explicitly how they involved members of the target group and communities to enhance the feasibility of the intervention. Based on the descriptions in the articles, the different approaches to community involvement can be situated on a continuum of participatory approaches, ranging from a consultation role to a complete participatory process. We labelled these studies along this continuum: from community-informed (CI) over community-shaped (CS) interventions to community-driven (CD) initiatives (Attygalle, 2020), see Table 3. Community-informed studies consulted community members in the preparatory phase of intervention (Lovell et al., 2014), for translation of intervention materials (Ünlü Ince et al., 2013; Christodoulou et al., 2018) or made reference to applying a community-based intervention, but researchers maintained the full control of the intervention study. Community-shaped studies actively involved community members throughout the development or implementation of the intervention. Researchers worked in collaboration with community members to ensure the intervention's cultural appropriateness. This was done by Gater et al. (2010) as they developed the group activities in their intervention together with voluntary organisations. In other studies, intervention services were provided by community members, for example, community health workers delivering the intervention (Afuwape et al., 2010; Hesselink et al., 2012) or giving education sessions (Kocken et al., 2008). In the study of Chaudhry et al. (2009) female Urdu-speaking drivers picked up Pakistani women to ensure that family or community members did not object to the women going out alone with a taxi driver. Baskin et al. (2021) discuss in their scoping review

community-centred interventions, which are either implemented in community settings or in a health setting but delivered by the community members and/or the voluntary sector. In the specific study of Lwembe et al. (2017), the researcher was part of the intervention as participant observer to evaluate the use of a co-production approach to improve access to psychological therapies.

Community-driven interventions went a step further and ensured community participation from the start of the intervention development process until its evaluation. This is extensively elaborated in the research report by Edge et al. (2018) who co-developed a cultural adaptation of an existing family intervention in partnership with African-Caribbean service users, their families, community members and healthcare professionals. Using a different approach, Malone et al. (2017) developed, implemented, and evaluated an arts-based community intervention to create awareness of suicidality among Irish Travellers in collaboration with the population group throughout the entire research process. Mantovani et al. (2017) used a qualitative participatory approach to pilot an outreach intervention addressing the mental health needs of African and African-Caribbean groups, where faith-based organisations, local public services and community services co-produced the pilot project. De Freitas and Martin (2015) applied the framework of the Participation Chain Model to ensure minority user participation in a mental health advocacy project. In the "community conversation" intervention of Knifton et al. (2010), health and BME community organisations designed and delivered supportive workshops to explore mental health and stigma.

These studies demonstrated that community-based initiatives were promising approaches, and they were able to document positive changes in their envisaged outcomes as described above (see interventions' effectiveness).

### Barriers and facilitators to intervention uptake

Different barriers impeded the successful implementation of interventions. For instance, the heterogeneity across outcomes and target (sub)groups complicated the cultural adaptation of an intervention (Degnan et al., 2018; Eylem et al., 2021). Some authors mentioned that it was difficult to engage the "hard-to-reach" target groups for regular health interventions (Reijneveld et al., 2003; Ünlü Ince et al., 2013) and faced poor participation (Hesselink et al., 2012; Lovell et al., 2014). According to Lovell et al. (2014), recruitment to primary care trials in the United Kingdom was generally problematic and especially difficult in mental health trials. The complexities of migration-specific barriers complicate recruitment, as well as the development of culturally acceptable and accessible interventions (Lovell et al., 2014). The hindering effect of these external, social determinants were often described. Factors such as employment, financial difficulties, legal status, acculturation, racism, and discrimination, might have a large effect on migrants' and ethnic minorities' mental health status and therefore might reduce interventions' (long-term) effects (Osman et al., 2021). To a similar extent, social problems, many of which are connected to the family context, were brought up by the researchers, especially in those studies describing interventions targeting women. For instance, maltreatment by the husband, problems in raising their children or unavailability of childcare, disabled relatives or divorce were mentioned as impeding interventions' success (Kocken et al., 2008; Gater et al., 2010; Khan et al., 2019). Due to sociocultural norms, women may experience a lack of autonomy in movement

and decision-making, some women expressed that their husband would prevent them from participating in treatment (Khan et al., 2019). The resistance from family members was subject to the stigma on mental health and fear of anticipated disclosure of mental health problems to the "outside" world (Gater et al., 2010). Maintaining family honour and a need to keep up appearances within the community, hinders these target groups from participating and mental health problems are likely to be covered up (Khan et al., 2019). Mental health stigma was experienced as a major barrier among different target groups (Lwembe et al., 2017; Mantovani et al., 2017; Christodoulou et al., 2018), but at the same time attempted to be broken down by specific intervention studies, such as by Mantovani et al. (2017).

Community engagement emerged as a potential facilitator to engage people from target groups more easily (Lovell et al., 2014). People felt connected with a provider or with the intervention itself, when they were able to relate to the content of it, felt being listened to, and experienced their needs to be accommodated (Christodoulou et al., 2018; Eylem et al., 2021). This provided them with a sense of empowerment (Christodoulou et al., 2018), which might disrupt power balances and may give room to dialogical and equitable encounters (de Freitas and Martin, 2015). Making meticulous (cultural) adaptations to the contents and method of delivery to this target group is thus essential (Reijneveld et al., 2003). (Social) connectedness can be facilitated by cultural adaptation of intervention, such as adaptations to language, adaptations in the domains of concepts and illness models, cultural norms and practices, considering explanatory models of illness, incorporating spiritual/religious activities, and acknowledging culture-specific familial structures (Degnan et al., 2018), inclusion of narratives delivered by community service users (Knifton et al., 2010) or making organisation-specific cultural adaptations (Arundell et al., 2021). Another key factor in engaging and retaining participants was engagement with their families (Khan et al., 2019), and also group interventions were evaluated positively in creating this feeling of social connectedness (Lovell et al., 2014). This connectedness can be further enhanced through participatory approaches, by involving migrant health educators (Kocken et al., 2008), training lay health workers from the same community to deliver the intervention (Baskin et al., 2021), using expert patients, and giving ownership of intervention modalities or shared decision-making of stakeholders (Lwembe et al., 2017; Table 6).

### Discussion and conclusions

This scoping review mapped and synthesised studies of interventions designed to improve the mental health or mental well-being of migrants and ethnic minority groups living in Europe. Such

**Table 6.** Barriers to and facilitators for successful intervention uptake

| Barriers | Facilitators |
|---|---|
| • Heterogeneity of target (sub) groups and outcomes<br>• Difficulties in recruitment and participation of target groups because of<br> ○ Migration-specific barriers<br> ○ Socio-cultural norms<br>• External social determinants impeding long-term effects<br>• Mental health-related stigma | • Community engagement<br>• Cultural adaptation of content and method of delivery<br>• (Social) connectedness through family engagement or group interventions<br>• Participatory methodology approaches |

interventions are highly needed as these groups are at higher risk for mental health problems than Europe's general population (Carta et al., 2005; Marmot et al., 2010; Missinne and Bracke, 2012). Because of structural inequalities in society (Carta et al., 2005; Missinne and Bracke, 2012), stigma associated with mental health (Kocken et al., 2008), language barriers (Bhui and Bhugra, 2004), and different cultural perceptions of what may constitute mental health problems (i.e., differing explanatory models; Lovell et al., 2014), tailored approaches and interventions are required to reach these population groups. Our review shows that attention to meet these specific demands is increasing. Yet, given the result of the limited amount of only 27 studies over a period of 22 years, this calls for a greater investment in documenting mental health interventions. Within the selected studies, the effectiveness of some interventions is manifest, while other described interventions were too small-scale or in a pilot phase and need a larger and long-term implementation to evaluate their impact, which impedes us to draw general conclusions on effectiveness and scalability. However, our synthesis and analysis indicate a strong added value of specifically targeting migrants and ethnic minority groups. We identified successful intervention mechanisms to promote mental health in these populations, such as having a sound theory-base, culturally adapting evidence-based interventions, or applying a participatory approach during the development/adaptation of targeted interventions. In what follows, we first critically discuss the findings of our research objectives: (1) the available interventions and their respective outcomes, (2) the intervention mechanisms and cultural adaptation and participatory strategies used, as well as (3) barriers and facilitators for intervention uptake. We then point out the limitations of the selected studies, as well as of our scoping review. Lastly, based on our findings, we map out recommendations for future research, policy, and practice.

### *Available interventions and outcomes*

The findings of this scoping review contribute to an increasing evidence-base of effective mental health interventions. More than half of the studies reported statistically significant results for their envisaged outcomes. If no significant results were found, relevant precursors to improve mental health were identified. Six out of seven randomised trial studies showed positive effects on mental health outcomes, mostly a reduction in depression rates (Jacob et al., 2002; Reijneveld et al., 2003; Kocken et al., 2008; Afuwape et al., 2010; Osman et al., 2017; Siddiqui et al., 2019). Also, studies acknowledging the difficulty to conduct RCTs under complex real-life conditions, were able to demonstrate positive effects (e.g., Hesselink et al., 2012). Furthermore, interventions targeting the social and environmental circumstances in which they were implemented illustrated positive effects. An intervention aimed at changing social relations by improving parent–child interactions positively impacted the mental health of the parents (e.g., Osman et al., 2017, 2021). Lifestyle interventions promoting and facilitating access to physical exercise (e.g., Reijneveld et al., 2003; Rabiee et al., 2015; Siddiqui et al., 2019) proved to have a positive impact on mental health outcomes. Many smaller and/or qualitative studies indicated promising results or delivered insights into enabling factors for improving mental health and well-being such as improving social functioning (e.g., Hesselink et al., 2012) or increasing knowledge on mental health conditions (e.g., Gater et al., 2010) or direct support from community members or migrant health educators (e.g., Afuwape et al., 2010).

### *Intervention mechanisms, cultural adaptation, and participatory strategies*

Not all studies systematically shared information on their assumed intervention mechanisms. Only few studies described systematically, which (set of) changes were effective, how participatory approaches were exactly implemented, or which preconditions were needed to be in place to successfully implement the interventions. We thus cannot identify uniform mechanisms leading to potential effectiveness. However, this review reveals three principles that may increase an intervention's success: having a sound theory-base, making cultural adaptations, and using participatory approaches. Firstly, our findings show that using a sound theory-base seemed to yield better results (De Silva et al., 2014). The choice for a specific theoretical underpinning determined the choice for respective interventions strategies or elements, such as using CBT elements (Ünlü Ince et al., 2013), peer-support or participatory strategies to increase empowerment (e.g., de Freitas and Martin, 2015). Making the underlying theory base explicit, made it easier to assess an intervention's effect. In addition, it also makes studies replicable, which is an important consideration for further increasing the evidence base. A thorough consideration of an explicit theory of change can contribute to the sustainability of an intervention's beneficial effects (Mayne, 2020). By grouping the identified intervention mechanisms according to the level of social determinants of health they addressed (see Table 4), we made an effort to fill the gap identified in the literature that current classification systems of mental health interventions do not inherently address health inequities relevant for migrant mental health, such as health disparities based on, for example, ethnic inequities or socio-economic status.

Secondly, cultural adaptations of an intervention increase its acceptability for several reasons. Cultural adaptations enhance the perceived social connectedness among study participants and facilitate access and familiarity with an intervention through the use of cultural reference systems and appropriate language. They contribute to reducing stigma associated with mental health and interventions. Furthermore, culturally adapted interventions consider the specific structural conditions in which many migrants and ethnic minorities live. The level of cultural adaptation among the reviewed studies ranged from translation of materials (Eylem et al., 2021), over adapting the content of the intervention by including cultural idioms and metaphors (Khan et al., 2019) to broadening the implementation of the intervention by addressing socio-economic circumstances (Rabiee et al., 2015). Drawing general conclusions to what extent cultural adaptations should be made and what their nature should be, is challenging given the diversity of the included studies. However, a one-size-fits-all approach may not address the diverse needs of different migrant and ethnic minority populations. However, involving communities affected may help to tailor the needed services.

Thirdly, the evidence presented here shows that community involvement and participatory strategies – preferably starting from the conceptualisation of the (or: adaptation of) the intervention – facilitate its development and successful implementation. Actively involving the communities not only contributes to solely adapting the intervention itself, but also creates cohesion and empowerment among the participants. Studies showed that including patients and their families as well as community leaders can be essential to overcome recruitment barriers and low participation rates (Lovell et al., 2014; Dowrick et al., 2016; Lwembe et al., 2017; Edge and Grey, 2018; Riza et al., 2020; Baskin et al., 2021). Reasons for

reluctance to participate in interventions, such as voluntary nature, stigma, trust issues or lack of appreciation may hinder achieving an intervention's goals (Apers et al., 2021). The extent to which the interventions described in the selected studies were participatory varied considerably: from community-informed where community members were consulted in a preparatory phase (e.g., Lovell et al., 2014), over community-shaped interventions (e.g., Gater et al., 2010), to community-driven participatory approaches as the highest form of involvement towards creating community ownership (e.g., Edge et al., 2018). Studies showed that the use of participatory approaches increased effectiveness as it improved social connectedness, power in local decision-making, and even structural issues such as access to safe and affordable housing (Baskin et al., 2021).

Our findings show that in addition to the above-mentioned principles, it is equally important to address the broader socio-ecological community context. Applying a holistic and intersectoral approach is needed, in particular for people and groups living in precarious socio-economic conditions. (Clinical) interventions or treatments for a general audience (e.g., Kocken et al., 2008) have encountered difficulties to address these social determinants. Additionally, financial implications cannot be neglected, calling for free or payable access to health promotion activities. Intervention uptake, however, is facilitated when socioeconomic conditions are taken into account, and social contact and cohesion are enhanced (e.g., Rabiee et al., 2015).

### *Facilitators and barriers for intervention uptake*

Applying (a combination of) these intervention mechanisms is thus likely to increase the feasibility and acceptability of mental health interventions among the intended target groups. These strategies contribute to overcoming barriers related to the stigma on mental health complaints by incorporating diverse explanatory models in healthcare (e.g., Lwembe et al., 2017; Mantovani et al., 2017; Christodoulou et al., 2018) or through the co-production of interventions with migrant and ethnic minority communities (e.g., Lwembe et al., 2017; Edge et al., 2018), for instance using artistic approaches (Van De Venter and Buller, 2015; Malone et al., 2017). Also, the holistic nature of many mental health complaints, related to the structural living conditions in which ethnic minorities and migrants live, should be considered to a larger extent (e.g., Knifton et al., 2010; Mantovani et al., 2017). However, hindrances related to these mechanisms should also be acknowledged. Again, the observed heterogeneity of interventions and target groups complicated defining which level of specificity cultural adaptations should have. While interventions often yielded successes in finding coping strategies to deal with mental health problems by introducing direct support from migrant health professionals, it seemed to be far more difficult to increase social support from participants' direct personal environment (Kocken et al., 2008). External, socio-ecological factors, as well as other social problems such as stigma on mental health from the near social environment created resistance to participation. Existing notions on how to engage social networks for all participants were challenged (Chaudhry et al., 2009).

### *Limitations*

#### *Limitations of selected studies*

The broad range of described mental health outcomes (e.g., from reducing mental health stigma to reducing suicidal risk) challenges a comparison across studies. While our review identified several examples of pilot and small-scale studies with promising results,

many studies were underpowered to assess the envisaged outcomes. These studies should be replicated and evaluated in larger studies using rigorous designs to deliver the needed evidence for upscaling. Additionally, a potential hindrance to scaling up may be the difficulties in the recruitment of participants encountered in several of the intervention studies (e.g., Knifton et al., 2010; Lovell et al., 2014; Van De Venter and Buller, 2015; Christodoulou et al., 2018). Authors mentioned difficulties in defining and reaching the target groups, and reported poor participation (Hesselink et al., 2012; Lovell et al., 2014). This resulted in relatively small sample sizes, short follow-up periods and lack of statistical power. Another review on a similar topic identified transcultural barriers to participation in early intervention research for migrants and ethnic minorities clearly showing a selection bias (e.g., Deriu et al., 2018). The selected studies also did not entail cost-effectiveness measures (with the exception of Afuwape et al., 2010), yet, cost-effectiveness could be the most convincing argument for policy-makers for scaling-up effective interventions.

The selected articles were also very heterogeneous in terms of the study participants: target groups and inclusion criteria varied across study settings (i.e., specific recruitment criteria), and used different categorisation in different migration and socio-economic contexts. This complicates the comparability of the studies, making it difficult to draw conclusions across studies. Not paying sufficient attention to target-group specific characteristics could also result in overgeneralization (Eylem et al., 2021). Additionally, our scoping review revealed a geographic bias. Most studies took place in the United Kingdom followed by the Netherlands and Sweden, and only a few single studies were conducted in other European countries. This bias could be due to differences in investments in the prevention of mental health and addressing underlying health inequalities. For instance, the Equality Act (2010) in the UK might explain the large amount of UK studies, as it has promoted enquiry into ethnic inequalities (Iliffe et al., 2017) by explicitly focusing on conducting needs assessments, resource allocation, and health care planning (Acheson, 1998). This has positively impacted research opportunities (Mathur et al., 2014).

#### *Study limitations*

Some limitations of this scoping review should also be pointed out. Due to limited resources, we could not include grey literature, hence we might have missed practice-based evidence often reported by local authorities, civil society organisations and health care practitioners (WHO, 2015). Data extraction was done by single authors, and not in duplicate. In line with the methodology of a scoping review, we did not systematically assess the quality of the included studies, yet we critically reflected on their potential flaws. Furthermore, we opted for interventions in Europe only, however, these regions could also learn from the implementation of interventions outside Europe. Using the categorisation of the public health prevention continuum may also present a limitation. Some studies addressed overlapping stages within the continuum, for example, evidence-based treatment approaches combined with mental health promotion at community level (Eylem et al., 2021; Dubus, 2022). As indicated by Compton and Shim (2020), this framework does not inherently address health inequities relevant for migrant mental health. To mitigate this, we have synthesised the information available on intervention mechanisms according to the different levels of social health determinants. The broad definition of the population of interest, reflecting the lack of a fine-grained definition, made comparisons across studies difficult. Finally, there may be a bias due to the researchers' cultural influences and cultural

competency indicating the need to develop cultural protocols for researchers and strengthen researchers' cultural competency.

## Recommendations for research and policy

Based on our synthesis, despite the above-described challenges and study limitations, the following recommendations for research and policy can be made.

### Research recommendations

First, our scoping review illustrates that evidence on migrant and ethnic minority groups other than refugee and asylum-seekers is scarce and very diverse, few interventions measured the broader social change and reduction in health inequalities and only broadly referred to it in the discussion section. Broadening the scope of research towards other migrant populations would generate more insight into the (common) challenges that these groups face. This could help in addressing the underlying drivers, such as structural issues that affect the mental health of all migrants. Research should invest in studying and addressing the interplay of cultural specificities with socio-ecological factors. Mapping specific mental health outcomes, implemented interventions and outcomes per ethnic minority and migrant group, could work revealingly (Uphoff et al., 2020).

Second, our review shows little consistency in defining study populations when considering these "other migrant groups" (e.g., few studies differentiated between first- and second-generation migrants, studies were based on language spoken by participants, studies focussed on "hard-to-reach" migrant groups). To achieve better comparability across studies, future research will therefore benefit from a more streamlined and fine-grained definition of the different categories of migrant population groups, as well as the studies' geographical location. Studies should clearly specify how migrant and ethnic minority populations are defined, that is, per migration community, legal status, cultural and/or religious background. In doing so, future research could specify per migration community, drivers of migration and cultural similarities with the dominant country. This could further facilitate the understanding of and subsequent narrowing health inequality gaps in different contexts (Lebano et al., 2020; Van Apeldoorn et al., 2022).

Third, while stakeholder participation and intersectoral approaches are acknowledged, most interventions remain at the individual level. Participatory interventions, however, not only demand good communication, dealing with linguistic barriers and cultural differences, but also close collaboration of all stakeholders, ranging from governmental actors to ethnic minority and migrant communities (Riza et al., 2020). There are some promising studies – mostly qualitative accounts of participatory initiatives – to improve mental health outcomes among migrants and ethnic minorities in the EU/EE, but there is a paucity of high-quality evidence regarding these approaches. More studies should look into community-based or community-led approaches.

Fourth, many intervention studies insufficiently considered outcome measures that assess a holistic approach to mental health, and mainly focus on depression or anxiety scales to evaluate interventions. More in-depth analyses on the impact of the different intervention stages could be insightful and yield distinct mental health outcomes as well as the preconditions to successful implementation. In particular, intervention and implementation research would be needed in developing multi-level interventions, addressing both proximal and distal intervention outcomes. From a public health point of view, investments should be made in social and health systems research that addresses the quality of mental health interventions taking into account the sociocultural and socioeconomic contexts and approaches (Haro et al., 2014). Future research should systematically assess the specific preconditions for interventions to be effective and to reach the intended target group, as well as its cost-effectiveness (Afuwape et al., 2010).

Fifth, given the gender gaps in mental health complaints, a more gender-based approach in interventions are needed, and attention should be paid to how gender intersects with other social determinants of mental health and ethnicity (Bhugra, 2004; D'Souza and Garcia, 2004; Van De Venter and Buller, 2015; Baskin et al., 2021). Also, given the explicit focus in some studies on women (Jacob et al., 2002; Kocken et al., 2008; Chaudhry et al., 2009; Gater et al., 2010; Hesselink et al., 2012; Khan et al., 2019), for instance when focusing on post-natal and maternal related mental health (Hesselink et al., 2012; Khan et al., 2019), or depression (Gater et al., 2010), it would also be of added value to focus more on men. Understanding their specific explanatory models on mental health, perceived masculinity, and power dynamics in the household would give insights in potential barriers to mental health services. In turn, this could improve both men's and women's participation in interventions on mental health.

### Policy recommendations

Some policy recommendations can also be formulated. A holistic approach to mental health, considering the potentially differing explanatory models between healthcare practitioners and ethnic minority groups and migrants, as well as the socio-economic circumstances in which most migrants and ethnic minorities live, needs to be implemented in future interventions and policies. It is important that the social determinants defining ethnic minorities' and migrants' vulnerabilities to mental health problems are addressed. However, as we ranked the studies in Table 4 according to the levels of social determinants, we conclude that only few interventions target the resettlement stressors, such as socio-economic and living circumstances, legal residence procedures, detention procedures, and experiences of discrimination and racism (Priebe et al., 2016; Lindert et al., 2008; Nosè et al., 2017; Von Werthern et al., 2018). This might be explained by the fact that structural factors are difficult to change and might require structural policy adaptations, such as an intersectoral health-in-all approach (WHO, 2015). However, addressing the societal context is crucial for prevention strategies, as it is often where the mental health challenges that these groups are confronted with arise (Marsella, 2011). This requires a holistic and interdisciplinary approach, involving governmental actors who have the power to influence harmful structural factors. Hence, policy makers should invest in efforts to streamline services so they fit and interact better together to facilitate the implementation of a holistic approach. Some authors argued (Arundell et al., 2021) that culturally adapted care is needed, in which all interventions, services and treatments are suited for different cultural values, patterns, behaviours and beliefs. This could start with a better representation of ethnic diversity within the healthcare systems, as noted by Baskin et al. (2021).

Furthermore, needs assessment, tailored health care planning and resource allocation (Acheson, 1998) could be facilitated by the registration of ethnicity, and migration background (Lebano et al., 2020; Van Apeldoorn et al., 2022). This way, the needs of local ethnic minority and migrant groups, as well as the specific risks and needs concerning mental health, could be better assessed to make healthcare culturally adapted (Arundell et al., 2021), and to foresee

some targeted additional interventions oriented at reducing mental health complaints of ethnic minority and migrant groups. Finally, medical interventions, healthcare systems and practices undergo constant transformations, such as digital transformations in communication, assessment, and follow-up. More attention should be paid to a still existing digital divide for migrants and ethnic minorities, which COVID-19 has revealed (Nöstlinger et al., 2022). More insights are needed into accessibility, cultural attitudes, and migration-specific experiences of such digital tools (Marwaha and Kvedar, 2021).

**Open peer review.** To view the open peer review materials for this article, please visit http://doi.org/10.1017/gmh.2023.15.

**Supplementary material.** The supplementary material for this article can be found at http://doi.org/10.1017/gmh.2023.15.

**Data availability statement.** All articles are publicly available in the described databases and an appendix of Supplementary Materials is available online.

**Acknowledgements.** The authors thank the Global Mental Health editors' team for their assistance throughout the writing and publication process.

**Author contribution.** C.A. and H.A. defined the scope of the review. H.A., L.V.P., and C.N. determined the review methodology, conducted the review, and wrote a first version of the manuscript, which was revised by C.A. All authors reviewed the final manuscript.

**Financial support.** The authors received no financial support for the research, authorship, and publication of this article.

**Competing interest.** The authors declare no conflict of interest with respect to the research, authorship, and publication of this article.

**Ethics standard.** The protocol of the study has been registered at the Center for Open Science (https://doi.org/10.17605/OSF.IO/R8SBF).

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
