## [Reviewer Report]

*Comments to Author*: Background: The background section is well written with up to date references. However, it will be useful to talk about the economic burden of mental health problems in migrants and ethnic minority groups.

Methods: Multistage methodological framework to do this scoping review seems useful as this has helped to clearly articulate each step involved in review process.

In Stage 2 – Identifying relevant studies, it will be useful to add key search terms that made search string that was used for search run.

The strength of the analysis process was that both authors screened all the articles that added to the rigor in analysis.

In stage 4 - Charting of the data, it will be useful to know whether the extracted data was reviewed or cross checked by an independent person?

Only 27 studies in 22 years clearly indicate need for more research studies

On page 6 where authors talked about the types of information they extracted regarding interventions, authors may consider mentioning that they also extracted information about whether included studies made cultural adaptations in the intervention to make it more relevant for the specific groups (migrants/ethnic minorities). As later on page 8 they clearly explained that adaptations were made in interventions in some included studies.

Results: In results section, authors have very comprehensively explained studies’ characteristics, intervention types using prevention continuum, pathways to successful intervention uptake by highlighting the theory-base and intervention mechanisms and also analyzed the barriers and facilitators for intervention uptake.

Discussion: in discussion section where author talked about socio-ecological factors as potential barrier in intervention uptake, author may consider researchers’ cultural influences and researchers’ cultural competency. Therefore, in order to overcome cultural issues, other than community engagement and involvement it would be useful to prioritize developing cultural protocols for researchers and strengthening researchers’ cultural competency

Authors may consider implications of findings of this review for ethnic minorities in LMICs, as if there is lack of investments in health inequalities in HICs such as UK, the situation is even worse in LMICs. Therefore, authors may discuss how these results could inform adaptations to be made in interventions for mental health problems in this population in LMICs.

---

## [Reviewer Report]

*Comments to Author*: I found this paper well written and the methods clearly described. This is (however) a very large piece of work, which presumably aims to be a starting point for any research into mental health (including wellbeing) interventions in refugees in high income countries. I wonder if a scoping review on this topic is necessarily too large to be useful, and whether the authors should consider e.g. effectiveness, acceptability, scalability as separate questions- coudl the authors respond to this? The breadth of ambition here seems to lead to specific problems in my view- e.g. it’s not clear what outcomes mean here- whether this refers to effectiveness outcomes or a broader definition of outcomes(e.g. delivery/process outcomes). A more minor point is: the authors state that they piloted their selection/extraction strategies- could they clarify whether they made any amendments based on this?

---

## [Reviewer Report]

*Comments to Author*: Thank you very much for submitting this very interesting review. I think this is a promising paper, but as you can see from the reviewers' and my own comments below, I think major revisions would help strengthen the paper.

Abstract

- I am not sure if all the variables listed under social determinants really fit that classification. For example, I would see explanatory models as (culturally influenced) ways of thinking about mental health concerns and their causes, not as a social determinant

- A key conclusion from the paper (the need to more holistically address social determinants through multi-sectoral programming (e.g., addressing social exclusion and socio-economic adversity) appears to be missing from the abstract

Introduction

- Line 12: the post-migration context I think is not one social determinant, but a category encompassing a whole range of social determinants

- Line 20: I am not familiar with the term socio-ecological determinants of mental health. I see the term socio-ecological used more to speak about different levels at which social determinants can be analyzed (following Bronfenbrenner’s work)

- I agree with the recommendation of reviewer 2 that the scope of the paper provides a challenge, in that there is not enough space to go in-depth into some of the findings and recommendations. I would suggest to focus the paper on promotion/ prevention alone or treatment alone, and not the combination. Given the multiple existing (and recent) reviews in the treatment field, I think highlighting the promotion/ prevention studies would be the most interesting. It looks like there would still be enough papers for a substantial review if treatment studies are excluded. (This is hard to judge, because treatment studies are not separately discussed in the results section)

Methodology

- I appreciate the rigor in application of scoping review methodology, and the clarity of the 3 research objectives. 

- Population: it is not clear to me whether there were any restrictions on the country where participants migrated from (e.g., in terms of cultural background, or country income status). E.g., were studies with UK migrants in Switzerland, or Canadian migrants in Poland included?

- Line 93: what is English contingent?

- I did not see a convincing rationale for including studies from 2000 onwards. This seems a rather arbitrary cut-off

- Line 100: PsycInfo, not PsychInfo

- Line 114: ‘the’ missing before ‘following key characteristics’

- Cultural adaptation was included as a specific research objective, and is included under a separate paragraph under results, but is missing as a category in the data extraction sheet. How were data regarding cultural adaptation summarized?

- It appears data extraction was done by single authors, and not in duplicate. I think it would be good to mention that as a potential limitation in the discussion

- An important question for me concerns whether any appraisal of risk of bias was conducted for all of the included studies. I think this is usually a key element of systematic reviews, and would help disentangle the findings with high confidence from those with lower confidence

Results:

- Line 175: One qualitative study included forced migrants. In my mind that overlaps with refugees and asylum seekers, which were going to be excluded

- I think it would be helpful to provide an overview of the reasons why different migrant groups migrated (e.g., part of colonial histories, labor migration, etc). That 

- A key concern in my opinion is the categorization of different intervention types. First, I believe the primary, secondary, tertiary categorization labels are seen in the prevention field as slightly outdated. I think it would be more helpful to use the universal, selective, indicated prevention labels (e.g., as presented in the seminal Institute of Medicine reports from 1994 and 2009). Second, it looks like treatment interventions are conflated with tertiary prevention approaches (which was indeed a key concern with the primary, secondary, tertiary prevention labels). I think it will be helpful to exclude treatment-focused articles (as well as interventions aimed at increasing access to treatment), to allow a deeper discussion of the hypothesized mechanisms of prevention programs, which I imagine would be quite different from treatment mechanisms.

- Could Table 2 be presented by different intervention category? I think it would help engagement with the table. It would be very helpful if risk of bias assessment information would be included here.

- I think the section and table summarizing outcome measures could be deleted without a critical loss of information

- In my opinion, the section summarizing hypothesized mechanisms of interventions and theories of change is one of the most interesting in the paper (going beyond a simple summary of results from intervention studies, and filling a critical gap in the literature). However, because of the large scope of the paper at the moment, this section is not as developed as I think would be possible. For example, an overview figure (or table) highlight the key variables and processes (e.g., social determinants) addressed in interventions I think would be a highly valuable contribution to the literature. The second paragraph in this section in which CBT-based treatments are described falls a bit out of place, which I think argues for exclusion of treatment-focused studies from the review

- I think a short table to summarize the cultural adaptation strategies, focus and results of adaptation would be helpful to guide the reader through the narrative. 

- The labelling of participatory approaches seems helpful to me. I think it would be good to add this labeling activity in the methods section

- Line 454 Table X refers to which table? If table 4 is deleted, I think a short table summarizing participatory approaches and some key results would be helpful for the reader to have an overview while going through the narrative

- A short table summarizing barriers and facilitators would be helpful

Discussion

- For the first conclusion, it would seem to me that cultural adaptation would assist in improving an intervention for a range of reasons, not just by improving social connectedness within migrant populations

- Overall, I feel that the discussion section provides a helpful summary of the results and relevant recommendations. I wonder if the discussion section could be structured based on the 3 main research questions as paragraph headings. Currently, the discussion section seems to miss, for example, the research aim of identifying which interventions are available and their outcomes – and focuses mainly on the cultural adaptation and barriers/facilitators aims.

---

## [Reviewer Report]

*Comments to Author*: Thank you for attending to my comments on the earlier version. I hope you find my further comments helpful.

This is an important topic and it is to be hoped that this overview can helpful for the field. While I appreciate that this article was Editorially commissioned and the topic is broad, I did find the submission extremely long for a journal scoping review and wonder if it could be edited down in any way? I appreciate the difficulty here, given you have chosen not to narrow the scope despite an earlier recommendation about this - indeed your amendments seem to have increased the size and I would say, made the work less user-friendly.

Specific comments:

I would suggest that some reference to uncertainty/gap being addressed by the review be included in the abstract background, so that the reader can understand what the need is which is being addressed by the submitted work.

At the sentence beginning on line 9, it is unclear whether this is a finding/synthetic point emerging from the review- e.g. beginning the sentence with “We found that....” might be a useful way of flagging up that you are moving to the results(given you don’t have abstract subheadings).

There remains a bit of inconsistency between the abstract aim- “to review specific, targeted interventions on mental health promotion, prevention, and treatment” and the revised abstract finding that “interventions showed a positive effect on participants’ mental health, and indicated the importance of using a tailored approach.”. It might be clearer to specify “mental health interventions including health promotion, prevention, and treatment strategies....”, so that the meaning is clear.

Could you clarify throughout whether you are referring to “mental health disorders” or to mental disorders? I think mental disorders is more usual, and making this uniform throughout the paper would help differentiate from the non-diagnostic MH outcomes you have included.

I agree with another reviewer who suggests greater integration of social determinants theory into the introduction, but the changes you have made seem to imply SDoH are only relevant after migration, and not also determining the pre-migration context(access to employment, physical safety, educational opportunities).

Similarly, i think there is too limited an engagement with the particularly complex concept of inter-generational trauma transmission- this needs defining and expanding(line 41).

And again, some sense of the geopolitical context in which migration occurs seems to be lacking, when actually, the role of regional conflict patterns, economic disruption, state racism, seem to drive migration and are also relevant to the psychosocial context of individual migrants- e.g. intimate partner violence is more common in conflict settings.

Generally the introduction feels quite focused on the vulnerabilities of migrants- at the brief and slightly dismissive discussion of the healthy migrant effect at line 35, might the authors point to any resilience factors that migrants might have, even if evidence is limited - not all migrants develop mental health conditions, but I think the introduction reads as slightly one-sided in this respect.

Again, I appreciate the addition of reference to sociocentric/egocentric societies, but this reads as far too simple a reading of the evidence- how good is the evidence that this distinction a. exists across time, b. at a national population level(which is what is implied by the migration narrative at line 51), and c. the relation if any with mental disorder. So I would put this much more tentatively, e.g. “it has been proposed that populations vary according to a egocentric- sociocentric continuum(ref),...”.

Line 53-58: you refer to asylum seekers and refugees as a specific subgroup within migrants, but I think it would be helpful to state who is being excluded through this focus? i.e presumably economic migrants? And therefore subsequently, when you say at line 66:“ no review can be found....” do you mean on unselected migrant samples, or in (something like) non-refugee migrant populations? I think this could be clarified, and perhaps you could amend the aim at line 69 to state that you are going to focus on migrant populations as a whole, and exclude studies which focus only on refugees. Along the same lines, “In the current literature, no review can be found on mental health interventions for migrants and ethnic minority populations in Europe.”, may not be true if there are reviews on asylum seekers/refugees in Europe? The passage line 53-58 also needs expanding because it is the apparent justification for reviewing migrant mental health interventions and excluding asylum/refugee research studies- exactly what are the differences in needs experienced by asylum seekers (and compared to whom?); given you strongly stress the negative impact of migration on MH(as I refer to above), does that not make the case FOR studying this group even stronger. So the approach to exclude this group of studies needs to be much clearer, and contextualised to the broader migrant population and their possibly specific needs. Much of the same can be said in relation to focusing on adults and not children, especially given the reference to transgenerational transmission of trauma. This needs justification.

Sentence beginning at line 73 is incorrectly written and should be corrected.

“available to them” at line 84 is incorrect, and I don’t think is needed.

Please could you align the question at line 103 with the abstract and with the last lines of the introduction? And as above consider whether you need add a clarification because this isn’t the question you are actually answering(because you are excluding some studies on migrants, i.e. refugees/asylum seekers).

I gather from your response that databases were searched from their inception, so please could this be added to your description of stage 2? At results, you should state the overall period of study from which you drew the articles, e.g. if you are saying you searched e.g. pubmed from 1958-2022, you should state this, and say relevant articles came from 2002 onwards(as this is probably important?). And then at line 585 I am not sure whether this is correct- were papers from e.g. 1990 eligible or not(were they to exist)?

The sentence at line 163 is incorrect and should be rewritten. Which authors are being referred to at line 161?

At line 211- can you clarify whether or not you excluded studies specifically of asylum seekers and refugees?

Although I am not sure if footnotes are in style, I think the point being made in the footnote after line 217 should be in the main text and justified; why include the same programme twice(if the focus is on programmes)?

I am left confused about the inclusion of meta-analyses here. Could the authors clarify why studies contributing to the meta-analyses were not included, while the meta-analyses themselves were? Surely the individual contributing studies are as important for scoping?

The added table at line 429 is ok, but I think the description of underlying mechanisms for physical activity should be made more tentative and specific- i.e. which neurotransmitters are you talking about, and is the evidence for this sufficient.

I would replace undeniable at line 587 and anywhere else, given all evaluation involves some uncertainty.

I am not sure why the limitations of the included studies are reported separately from the rest of the results? I think incorporating the “limitations of selected studies” into the result would help to reduce the length of this paper which is already far too long for scoping review, in my view.

Could you include a reference for the levels of social determinants that you used to look at mechanisms?

Re. the recommendations- you should state the observations you made from your review, and then the ensuing recommendations, not the other way round. At line 741- could you connect this recommendation to something that you actually found in your review, e.g. the number of studies which did not differentiate 1st and 2nd gen migrants? And of course some of this lack of fine grained definition is by your design, as you excluded those studies which did actually focus on relatively homogenous migrant groups based on asylum status, I think?

I don’t understand how the equality act is relevant to research in Europe, perhaps higher level policy on racial and other equalities is relevant e.g. at UN or WHO levels?

Similarly I’m not sure how the sentence at line 752 emerges from your results, given you didn’t focus on these countries.

---

## [Reviewer Report]

*Comments to Author*: Interventions to improve the mental health or mental well-being of migrants and ethnic minority groups in Europe: a scoping review

GMH-22-0237.R1

Authors have successfully incorporated changes throughout the manuscript. However, following are some more comments that can help further strengthen the manuscript.

In first paragraph in “introduction” where authors talked about vulnerability of immigrant population to develop mental health problems can be further strengthen by a systematic review stating tenfold risk of mental disorders in this population compared to the majority

Fazel M, Wheeler J, Danesh J. Prevalence of serious mental disorder in 7000 refugees resettled in western countries: a systematic review. Lancet. 2005;365(9467):1309–1314. doi: 10.1016/S0140-6736(05)61027-6

Ekeberg, K. A., & Abebe, D. S. (2021). Mental disorders among young adults of immigrant background: a nationwide register study in Norway. Social psychiatry and psychiatric epidemiology, 56, 953-962.

Author may consider talking about healthcare utilization in migrants in UK and Europe;

Graetz V, Rechel B, Groot W, et al. Utilization of health care services by migrants in Europe—a systematic literature review. Br Med Bull 2016; 121: 5–18

In discussion, authors may consider suggesting Theory of Change approaches making interventions more relevant for sub-set of population and can help overcoming barriers in uptake of interventions.

Mayne J (2020) Sustainability analysis of intervention benefits: a theory of change approach. Canadian Journal of Program Evaluation 35, 204–2021.

Addition of information on mechanisms and social determinants of interventions in revised manuscript is useful.

---

## [Reviewer Report]

Dear Editor, dear Reviewers,

We would like to thank you for reviewing our revised manuscript. We have considered the feedback and have made changes in the manuscript as requested and where appropriate. We believe these changes have improved the manuscript. 

We trust that the changes are sufficient so that the manuscript can now be published.

Sincerely, 

Hanne Apers (on behalf of the co-authors)